# Loss of state transitions in Bryopsidales macroalgae and kleptoplastic sea slugs (Gastropoda, Sacoglossa)
Vesa Havurinne [1] ✉, Axelle Rivoallan [1], Heta Mattila[1], Esa Tyystjärvi [2], Paulo Cartaxana [1] & Sónia Cruz [1] ✉

Green macroalgae within the order Bryopsidales lack the fundamental photoprotective mechanisms of green algae, the xanthophyll cycle and energy-dependent dissipation of excess light. Here, by measuring chlorophyll fluorescence at 77 K after specific light treatments, we show that Bryopsidales algae also lack state transitions, another ubiquitous photoprotection mechanism present in other green algae. Certain Sacoglossa sea slugs can feed on Ulvophyceae algae, including some Bryopsidales, and steal chloroplasts – kleptoplasts – that remain functional inside the animal cells for months without the support of the algal nucleus. Our data reveal that the state transition capacity is not retained in the kleptoplasts of the sea slugs, and we provide evidence that the loss is caused by structural changes during their incorporation by the animals. Enforced chloroplast sphericity was observed in all studied kleptoplastic associations, and we propose that it is a fundamental property supporting long-term retention of kleptoplasts in photosynthetic sea slugs.

Kleptoplastic sea slugs are not content with simply eating their prey, macroalgae. These green sacoglossan animals steal chloroplasts from their algal prey and incorporate them inside their own cells as photosynthetically active foreign organelles, kleptoplasts, while degrading the algal nucleus and other components of the algal cytosol[1]. The peculiar kleptoplastic association between sea slugs and their algal prey has intrigued scientists for over a century[2–4]. Despite the interest, one of the main questions related to kleptoplasty remains open to this day: how are the kleptoplasts retained functionally for months inside an animal cell when being isolated from the gene products of the algal nucleus[5–7]?

In the light, the photosynthetic apparatus inside chloroplasts is constantly being damaged. This is largely due to the involvement of strong photosynthetic redox chemistry that can lead to excessive production of reactive oxygen species (ROS), particularly under bright light[8–11]. The repair and maintenance of the photosynthetic protein complexes, like the vulnerable Photosystem II (PSII), is thought to require tight coordination with the algal nucleus[12,13]. Furthermore, most of the thousands of proteins inside a chloroplast are in fact nucleus-encoded to begin with[14,15]. Most kleptoplastic sea slug species eat and steal chloroplasts from siphonaceous green macroalgae belonging to the clade Ulvophyceae[16,17], divided here to Bryopsidales and Ulvophyceae *sensu stricto* (or true ulvophytes; Fig. 1) following the classification by Hou et al.[18]. So far, sequencing the chloroplast genomes of these green macroalgae has not provided evidence to mark them

as being particularly extraordinary when it comes to their gene content, although retained genes encoding a chloroplast maintenance protease (*ftsH*) and a translation elongation factor (*tufA*) might imbue them with more autonomy than their contemporary plant or green algal chloroplasts[6,17,19,20].

Other avenues of investigation have focused on the common photoprotection mechanisms known to exist in green algae to explain why only chloroplasts from specific algal species are selected by the slugs for kleptoplasty[21–25]. Chloroplasts from some Bryopsidales algae, like *Codium* sp. or *Bryopsis* sp., can remain functional inside the sea slugs for over a month (Fig. 1)[26,27]. The selection of these algae for long-term kleptoplasty cannot be explained by an enhancement in non-photochemical quenching (NPQ) of excitation energy, a traditional photoprotective mechanism of green algae. Rather, the majority of Bryopsidales algae have been shown to lack qE, the energy-dependent quenching process induced by the protonation of the thylakoid lumen, and the xanthophyll cycle that modulates NPQ (see Supplementary Fig. 1 for a schematic), while these mechanisms have been shown to function in most other green algae, including the true ulvophyte macroalgae[28,29].

Further, even if the xanthophyll cycle and qE are present in the prey algae, they do not necessarily remain fully operational in the kleptoplasts. For example, in the generalist feeder sea slugs *Elysia viridis* and *Elysia crispata* feeding on *Chaetomorpha* sp. and *Acetabularia acetabulum*, respectively, the xanthophyll cycle switches on in high light, but it does not

[1]Department of Biology, CESAM—Centre for Environmental and Marine Studies, University of Aveiro, Aveiro, Portugal. [2]Molecular Plant Biology, University of Turku, Turku, Finland. ✉e-mail: vesa.havu@ua.pt; sonia.cruz@ua.pt

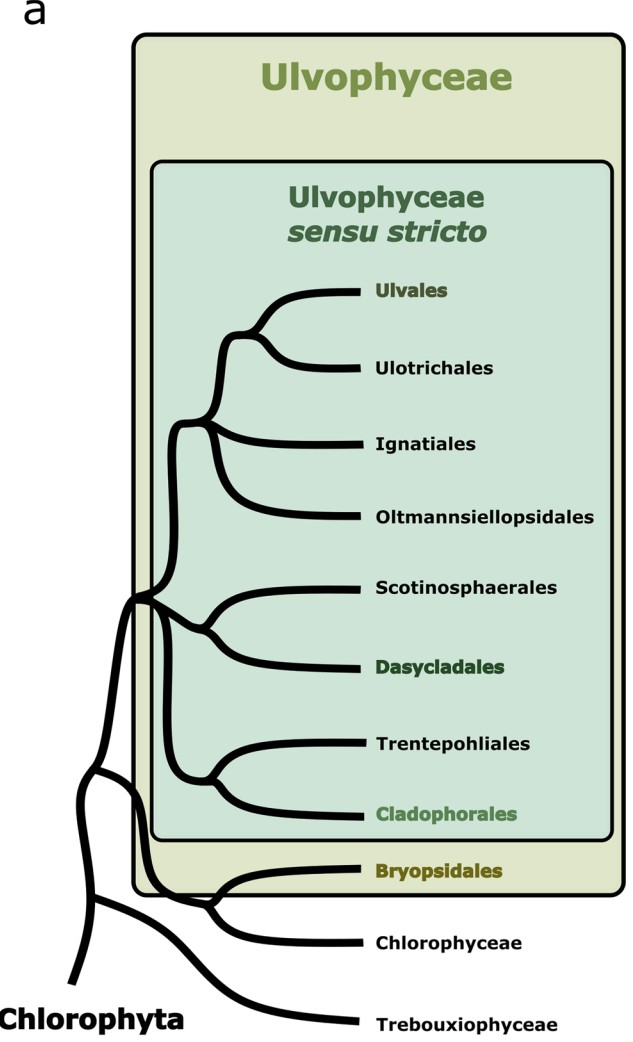

**Fig. 1 | The Ulvophyceae algae and the sacoglossan sea slugs feeding on them. a** A cladogram of Ulvophyceae showing the relationship between Bryopsidales and the core set of ulvophytes (Ulvophyceae *sensu stricto*), based on the classification by Hou et al.[18]. According to the traditional nomenclature, the class Ulvophyceae also includes Bryopsidales. **b** The sea slugs *Elysia timida*, *Elysia crispata*, and *Elysia viridis* are capable of long-term kleptoplasty. *E. timida* only feeds on and incorporates

chloroplasts from *Acetabularia acetabulum* as kleptoplasts, but *E. crispata* and *E. viridis* are known to obtain kleptoplasts from multiple sources, including algae belonging to the group Bryopsidales. Here, only the algae-sea slug associations tested in this study are shown. Like *E. viridis*, *Placida dendritica* also feeds on *Codium tomentosum*, but it only stores the kleptoplasts very transiently, and they are non-functional.

relax during subsequent darkness, like in the algae[24,25]. In the specialist feeder species *Elysia timida*, on the other hand, these photoprotective mechanisms of *A. acetabulum* remain fully operational inside the sea slug host[25]. The possible advantage of having kleptoplasts capable of qE and xanthophyll cycle over kleptoplasts originating from Bryopsidales has been difficult to assess, and comparisons between kleptoplasts deriving from different algae among sea slug species have led to contrasting views on their importance[22,23]. However, a recent study comparing the longevity of kleptoplasts originating from different algae in *E. crispata* revealed that *A. acetabulum* chloroplasts, with innate qE and xanthophyll cycle, are retained almost double the time compared to Bryopsidales alga *Bryopsis plumosa* chloroplasts[27].

The innate photoprotective mechanisms of different algae are likely not the only reason for long-lasting kleptoplasts. Nevertheless, the study of these mechanisms is important, as it may also give clues on the types of changes the sea slugs impose on the kleptoplasts to maintain them functional outside the algal cell[30]. One common photoprotection or light acclimation mechanism of green algae that has not been studied in ulvophyte algae or kleptoplastic sea slugs is state transitions. State transitions balance light-energy between PSII and photosystem I (PSI) during spectral changes of incident light, with an induction time of minutes-to-tens of minutes (see

Supplementary Fig. 2 for a schematic)[31,32]. The induction of state transitions is linked to the redox state of the plastoquinone (PQ) pool, a reserve of thylakoidal electron carriers: light conditions that preferentially excite PSII over PSI lead to a reduced PQ pool, increasing plastoquinol (reduced form of PQ) binding to the $Q_o$ site of the cytochrome *b6f* complex. This activates the algal Stt7 kinase that phosphorylates the mobile light-harvesting complex of PSII (LHCII), leading to its dissociation from PSII. The dissociated LHCII then moves to serve PSI in light harvesting (state 2)[33–36]. Conversely, preferential excitation of PSI oxidizes the PQ pool and allows the PPH1/TAP38 phosphatase to dephosphorylate LHCII, causing its migration back to serve PSII (state 1)[37–39].

Here, we show that all the tested macroalgae in the class Ulvophyceae possess state transitions, except for the Bryopsidales. In accordance, our results also indicate that kleptoplastic sea slugs *E. viridis* and *E. crispata* feeding on Bryopsidales algae are also incapable of state transitions (see Fig. 1b for all algae and sea slugs used in the study). What was surprising, however, was the finding that the strong state transition capacity of the chloroplasts of the alga *A. acetabulum* does not survive incorporation by *E. crispata* or *E. timida*. We show that the intake of the chloroplasts by the sea slugs causes immediate reversal of state 2 to state 1 and imposes a highly

spherical structure in the acquired kleptoplasts. Similar loss of state transitions was also noted in *A. acetabulum* itself when exposed to salinity treatments that altered the structure of the chloroplasts, which provides a plausible mechanistic framework for the loss of state transitions in the sea slugs. Lastly, we demonstrate that the enforced sphericity of the kleptoplasts is a conserved trait shared by all tested long-term and short-term kleptoplastic sacoglossan sea slugs, but only long-term kleptoplastic species maintain the spherical shape and gain the photosynthetic benefits.

## Results

### Bryopsidales algae and kleptoplastic sea slugs are deficient in state transitions

State transitions were evaluated with low-temperature chlorophyll fluorescence measurements in several ulvophyte algae and kleptoplastic sea slugs. The samples were first exposed to 15 min of either red 660 nm light favoring PSII or 740 nm far-red light favoring PSI (photosynthetic photon flux density, PPFD, 50 µmol $m^{-2}$ $s^{-1}$ for both) to induce light acclimation states 2 and 1, respectively (see Fig. 2a for the treatment light spectra). After the light treatments, the samples were snap frozen in liquid nitrogen and ground to a fine powder, and diluted with quartz powder to prevent self-absorption. Chlorophyll fluorescence was measured at 77 K temperature using 450 nm blue light for excitation (see Supplementary Fig. 3 for the spectrum) to evaluate if the light treatments caused changes in the maximum fluorescence emission of PSI (~710–730 nm), normalized to maximum PSII fluorescence emission (680–690 nm).

The shapes of the fluorescence emission spectra at 77 K differed considerably among the tested algae, but most of the algae had clear emission maxima that fell in line with the above-mentioned known fluorescence emission ranges of PSII and PSI (Fig. 2). The only alga with spectral shapes deviating considerably from this norm was the Bryopsidales alga *B. plumosa*, which was seemingly missing any clear PSI fluorescence peak (Fig. 2d). The true ulvophyte alga *Ulva* sp. was also noteworthy, as it had up to 6 times more PSI fluorescence than PSII fluorescence (Fig. 2g). The 77 K chlorophyll fluorescence spectra measured from the sea slugs *E. viridis*, *E. crispata* and *E. timida* mostly resembled the spectra of their respective prey algae (Fig. 2b–e, j–l). However, *E. crispata* showed consistently higher fluorescence at around 695 nm than *B. plumosa* (Fig. 2d, e).

Surprisingly, there were no significant differences in the normalized PSI fluorescence between red and far-red light-treated algae belonging to Bryopsidales (Fig. 2b, d, f), suggesting that this monophyletic group of green algae does not have state transitions. Like their prey, Bryopsidales algae, *E. viridis* and *E. crispata* did not exhibit any signs of state transitions under the tested light treatments (Fig. 2c, e). In all true ulvophytes, the far-red light treatments resulted in a lower PSI/PSII fluorescence ratio than the red light treatments, indicating functional state transitions (Fig. 2g–j). In the sea slugs *E. crispata* and *E. timida* fed with *A. acetabulum* there were no signs of state transitions; regardless of the illumination, the fluorescence spectra in the sea slugs had the same characteristics as the spectra in the algae in state 1 (after the far-red light treatment) (Fig. 2k, l).

### Responses of true ulvophytes to PSII or PSI excitation are more variable than in Bryopsidales and sea slugs

To gain insights on the relationship between PQ pool redox state and state transitions in our samples, all ulvophyte algae and sea slugs were subjected to a light treatment sequence where the moderate intensity (PPFD 50 µmol $m^{-2}$ $s^{-1}$) light switched from red PSII-specific light to PSI-specific far-red light and back, all in 15 min intervals. The chlorophyll fluorescence parameter qL is an indicator of the fraction of PSII reaction centers in the open state, where the first stable electron acceptor in PSII, the plastoquinone $Q_A$, is oxidized, with correlation to the PQ pool redox state[40,41]. Both qL and NPQ were analyzed from the samples by saturating light pulse analyses with a pulse-amplitude modulation (PAM) fluorometer during the light treatment sequence (Fig. 3). The first red PSII light

caused qL to drop below 0.5 in all Bryopsidales, suggesting a reduction of $Q_A$ (qL values 1 and 0 indicate a highly oxidized and reduced $Q_A$, respectively), and there was only a slight increase in qL back to ~0.5 in *B. plumosa* and *C. tomentosum* during the rest of the first red light phase. Far-red illumination increased qL in all Bryopsidales, although slowly in *C. tomentosum* and *Caulerpa* sp., and not back to fully oxidized $Q_A$, whereas the final PSII-specific light treatment once again seemed to reduce $Q_A$ in a very similar way as during the first one (Fig. 3a).

Apart from the last red light treatment, the qL kinetics in the sea slug *E. viridis* containing *C. tomentosum* kleptoplasts were similar to the alga itself; after the first fast reduction of $Q_A$ at the onset of red light, qL showed a slow, continuous oxidation of $Q_A$ in both red and far-red light. However, qL was consistently at a higher level than in *C. tomentosum*, possibly suggesting a more oxidized $Q_A$ in the kleptoplasts (Fig. 3b). In *E. crispata* hosting *B. plumosa* kleptoplasts the qL parameter was relatively stable during the red light treatments and stayed at roughly similar levels as in *B. plumosa*, but $Q_A$ oxidation was slower in the sea slug in far-red light (Fig. 3b). NPQ induction in all Bryopsidales algae and sea slugs with Bryopsidales kleptoplasts was nearly negligible during the entire light sequence (Fig. 3c, d).

The $Q_A$ redox state was highly variable in the tested true ulvophyte algae species according to the qL parameter (Fig. 3e). The reduction of $Q_A$ was strongest in *A. acetabulum*, where qL was close to 0 in red light and only recovered to slightly above 0.5 during the far-red treatment. The qL of *A. acetabulum* was also by far the slowest to increase in far-red light compared to any other alga or sea slug. Contrary to *A. acetabulum*, *Ulva* sp. showed only very transient reduction of $Q_A$, if any, during the red light treatments. In *Cladophora* sp., qL stayed above 0.5, also suggesting a tendency to maintain $Q_A$ oxidized, but not to the same extent as *Ulva* sp. The qL kinetics in *Chaetomorpha* sp. were very straightforward (Fig. 3e) and resembled the kinetics of the Bryopsidales algae *B. plumosa* or *Caulerpa* sp. (Fig. 3a), especially under red light. Similar to *E. viridis* hosting *C. tomentosum* kleptoplasts, the qL data from the sea slugs *E. crispata* and *E. timida* fed with *A. acetabulum* showed that the animals may maintain more oxidized $Q_A$, and likely also the PQ pool, than the alga in all tested light treatments (Fig. 3f).

Since transition to state 2 requires a sufficient reduction of the PQ pool, it is possible that state 2 was not induced to a level noticeable with 77 K fluorescence measurements in *E. crispata* and *E. timida* (Fig. 2k, l) simply because the moderate intensity red light was not strong enough to reduce the PQ pool in the kleptoplasts (see Fig. 3e, f). The 15 min red light treatment was therefore repeated with *E. timida* using a higher intensity (PPFD 100 µmol $m^{-2}$ $s^{-1}$). However, the PSI fluorescence at 77 K was not significantly different (Supplementary Fig. 4; one-way ANOVA, *p* value = 0.0904, *n* = 3) from the lower-intensity red light or the far-red light treatments shown in Fig. 2l.

The moderate intensity light treatments induced very little NPQ in the true ulvophytes *Chaetomorpha* sp. and *Ulva* sp., but surprisingly, the PPFD 50 µmol $m^{-2}$ $s^{-1}$ red light was very efficient in eliciting a strong NPQ response in *Cladophora* sp. and especially in *A. acetabulum* (Fig. 3g). *E. timida* and *E. crispata* with *A. acetabulum* kleptoplasts had very similar NPQ kinetics to each other, but the NPQ levels were consistently higher in *E. timida* than in *E. crispata* (Fig. 3h), although not nearly as high as in *A. acetabulum*. NPQ induction was additionally tested in *A. acetabulum* and *E. timida* under white, blue (440 nm), green (540 nm), and red (625 nm) lights, all PPFD 100 µmol photons $m^{-2}$ $s^{-1}$ (Supplementary Fig. 5). Here, white and blue light induced higher levels of NPQ in *E. timida* than in *A. acetabulum*, whereas red and especially green light caused lower NPQ levels in the sea slugs than in the algae.

Except for *Chaetomorpha* sp., our data indicated a positive connection between $Q_A$ reduction and NPQ induction in all true ulvophytes that supposedly have qE, whereas similar levels of $Q_A$ reduction did not trigger NPQ in the Bryopsidales. Because of this, a separate experiment was conducted to determine the qE and xanthophyll cycle functionality of *Chaetomorpha* sp. The algae were exposed to a 60 min high-light treatment (blue

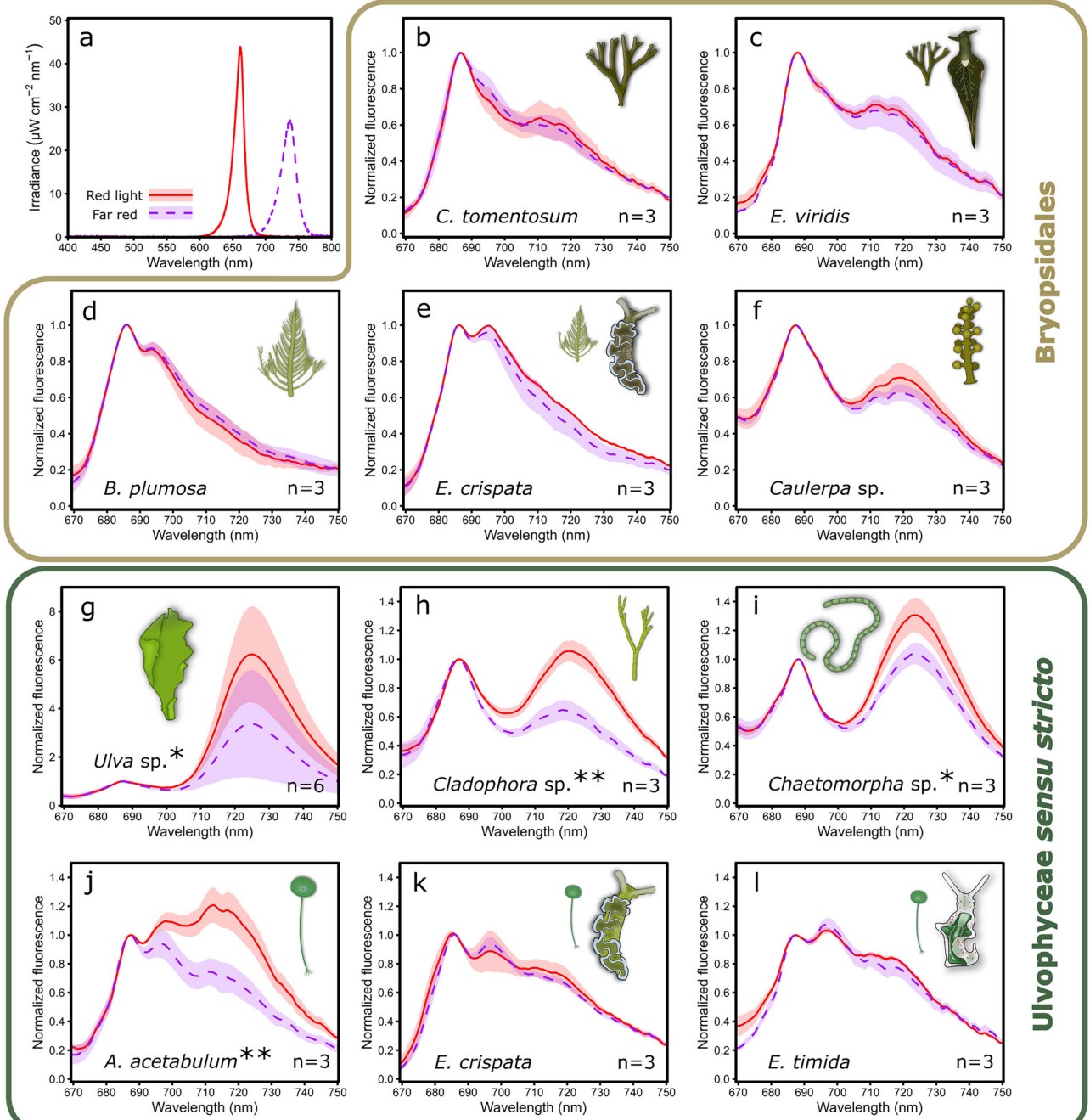

**Fig. 2 | State transitions in Ulvophyceae and kleptoplastic sea slugs with kleptoplasts originating from different ulvophyte algae. a** The spectra of the red PSII (red, solid line) and far-red PSI (purple, dashed line) specific lights used for the 15 min illumination (PPFD 50 µmol m$^{-2}$ s$^{-1}$) of the samples to induce light acclimation state 2 or state 1, respectively. **b–l** 77 K chlorophyll fluorescence of samples treated with red PSII (solid line) or far-red PSI light (dashed line). **b–f** Normalized 77 K fluorescence spectra in ulvophyte algae belonging to the Bryopsidales order (**b**, **d**, **f**) and in the sea slugs **c** *Elysia viridis* and **e** *Elysia crispata* possessing kleptoplasts derived from *Codium tomentosum* or *Bryopsis plumosa*, respectively. **g–l** Normalized 77 K fluorescence in ulvophyte algae belonging to **g** Ulvales, **h**, **i** Cladophorales, and **j** Dasycladales, as well as in the sea slugs **k** *E. crispata* and **l** *Elysia timida* that possess kleptoplasts derived from *Acetabularia acetabulum*. Excitation light was 450 nm in all fluorescence measurements. The number of biological replicates (*n*) is shown in the panels. The shaded areas around the curves show standard deviation. Significant differences in normalized PSI fluorescence between the light treatments are indicated with asterisks next to the species names (Student's *t*-test, *$p$ value < 0.05, ** < 0.01).

light; PPFD 500 µmol m$^{-2}$ s$^{-1}$) and a subsequent 60 min dark recovery period in the absence or presence of either 60 µM nigericin or 10 mM 1,4-dithiothreitol (DTT) (Supplementary Fig. 6). The addition of nigericin, an uncoupler that inhibits the formation of a proton gradient across the thylakoid membrane and therefore also qE formation, did not have any effect on NPQ induction, whereas DTT, a reducing agent that inhibits the conversion of violaxanthin to zeaxanthin, caused a clear decrease in NPQ

(Supplementary Fig. 6d). This suggests that *Chaetomorpha* sp. does rely on the xanthophyll cycle for NPQ, but not necessarily on qE.

## Kleptoplast incorporation overrides light acclimation state in *E. timida*

The dynamics of state transitions were investigated in more detail in the alga *A. acetabulum* by measuring 77 K fluorescence after different combinations

**Fig. 3 | Openness of PSII reaction centers (qL) and non-photochemical quenching (NPQ) kinetics in Ulvophyceae algae and kleptoplastic sea slugs during illumination with wavelengths favoring PSII and PSI. a, b** qL and **c, d** NPQ in Bryopsidales algae and the sea slugs *Elysia crispata* and *Elysia viridis* with Bryopsidales-derived kleptoplasts, as indicated in the legends. **e–h** The same parameters in true ulvophyte algae and the sea slugs *Elysia timida* and *E. crispata* containing kleptoplasts originating from the true ulvophyte *Acetabularia acetabulum*. The black, red, and purple bars on top of the top panels describe the low light (PPFD ~ 10 μmol m$^{-2}$ s$^{-1}$), red PSII light, and far-red PSI light treatments (PPFD 50 μmol m$^{-2}$ s$^{-1}$) during the experiments. For the spectra of PSII and PSI lights, see Fig. 2. qL and NPQ were determined by saturating light pulse analyses with a PAM fluorometer, and their definitions are described in the "Methods" section. The lines show the means from the biological replicates, and the colored areas around the curves indicate standard deviation. The number of biological replicates (*n*) is shown in the panels.

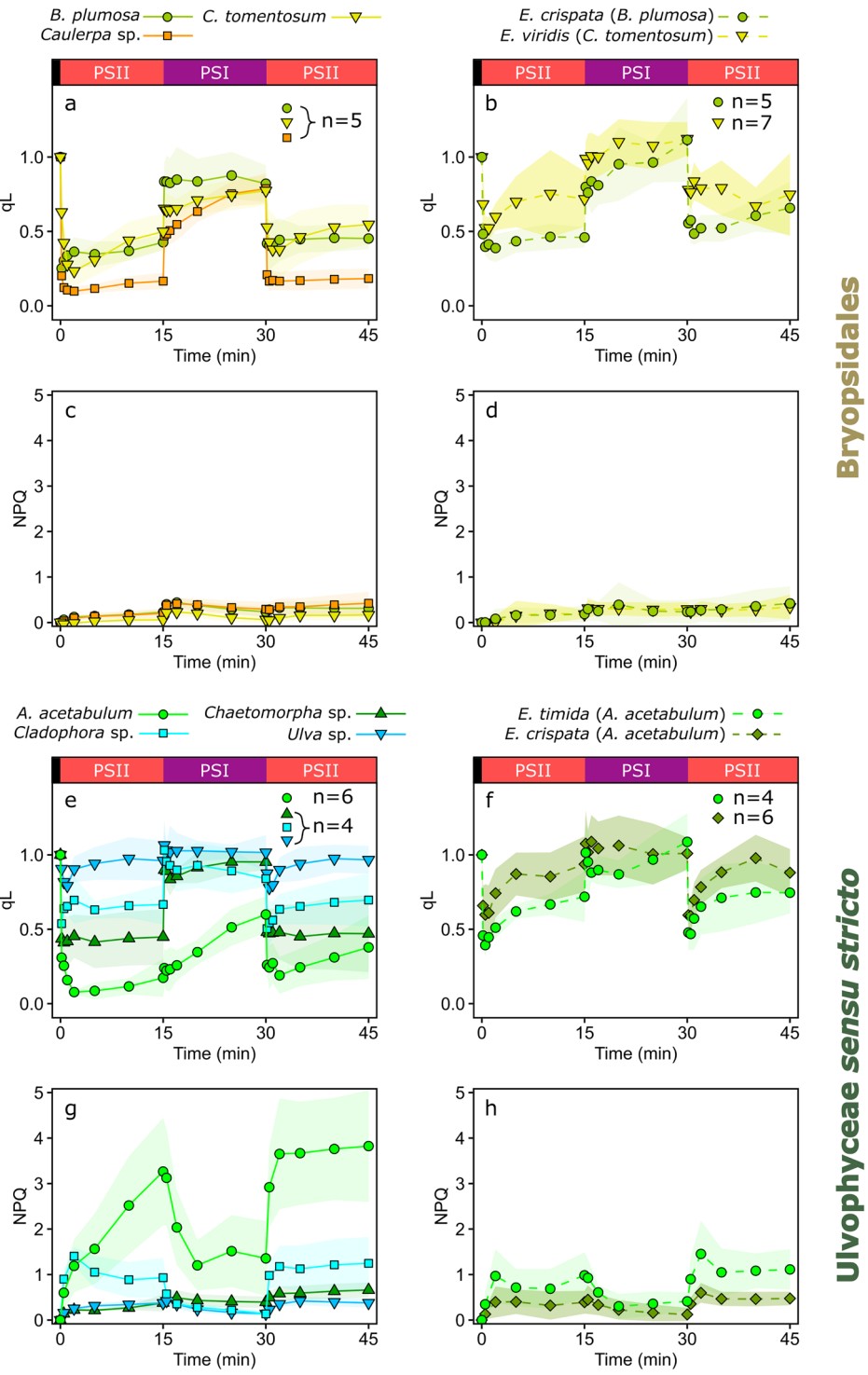

of darkness and PSII or PSI-specific light treatments (Fig. 4). Algae taken from growth conditions and then placed in the dark for 15 min showed an intermediate light acclimation state between states 1 and 2 (Fig. 4a). The treatment sequence of darkness, followed by 740 nm far-red PSI light and then 660 nm red PSII light, each one lasting for 15 min, resulted in state 2 (Fig. 4b) which, however, was not as strong as the state 2 reached by using only PSII light after darkness (Fig. 2j). When state 1 was first induced using far-red, and then the algae were dark acclimated again for 15 min, the algae remained in state 1 (Fig. 4c). Next, the algae were exposed first to red PSII light and then to far-red PSI light, which resulted in state 1 (Fig. 4d). The results show that reversal of the light acclimation states in *A. acetabulum*

falls within the regular minutes-to-tens of minutes time frame of state transitions. Finally, the addition of 5 mM sodium fluoride (NaF), a phosphatase inhibitor that prevents the dephosphorylation of LHCII, to the samples after 10 min of PSII light treatment inhibited the transition back to state 1 during a subsequent 15 min PSI light treatment (Fig. 4d).

The possibility of very strong dephosphorylation of LHCII as a cause for the permanent light acclimation state 1 in *E. timida* kleptoplasts was investigated next. For this, the sea slugs were first bleached from their chloroplasts with an 11-day period in high light (see Supplementary Fig. 7 for images of bleached *E. timida*), and the bleached sea slugs were then re-fed

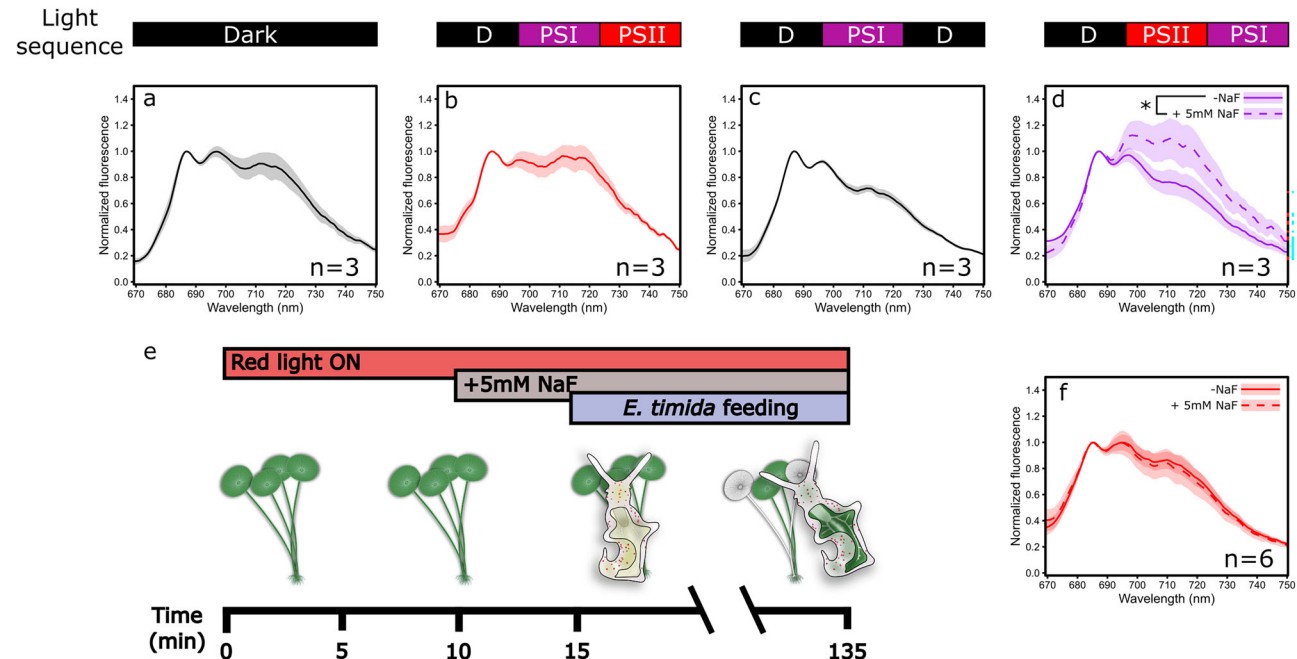

**Fig. 4 | Dynamics of state transitions in *Acetabularia acetabulum* and the forced light state 1 of *Elysia timida* kleptoplasts.** Chlorophyll fluorescence at 77 K measured from *A. acetabulum* samples that were **a** kept in the dark, **b** illuminated with 740 nm far-red PSI and 660 nm red PSII lights (PPFD 50 μmol m⁻² s⁻¹) after darkness, **c** kept in the dark, illuminated with PSI light and put to dark again, and **d** illuminated with PSII and PSI light in sequence after darkness ±5 mM sodium fluoride (NaF), added after the samples had been in PSII light for 10 min. The bars on top of the (**a–d**) indicate the light sequence the samples were exposed to prior to sampling. Each dark or light bar represents a 15 min period. **e** Schematic of the feeding experiment, where previously bleached *E. timida* individuals were allowed to feed on *A. acetabulum* cells pre-illuminated with PSII light for 15 min (to induce state 2) prior to releasing the bleached sea slugs to feed. To prevent dephosphorylation-dependent state 2 to 1 transition in half of the samples, 5 mM NaF was added after 10 min of the PSII light treatment, and the bleached sea slugs were added 5 min later to feed on the algae in the presence of NaF. Sea slugs that had turned noticeably green were sampled at different times throughout the 120-min feeding time. The sea slugs and the algae were continuously illuminated with red PSII light during the entire experiment. **f** 77 K chlorophyll fluorescence of *E. timida* from the experiment described in (**e**), ±NaF, as indicated. Excitation light was 450 nm in all fluorescence measurements. The lines show the mean and standard deviation is indicated by the shaded areas around the curves. The number of biological replicates (*n*) is shown in the panels. Significant differences in normalized PSI fluorescence between the ±NaF treatments are indicated with an asterisk (Student's *t*-test, *$p$ value < 0.05).

under red PSII light with *A. acetabulum* pre-illuminated for 15 min with the same red light, ensuring that the chloroplasts taken up by the sea slugs were in state 2. In addition, half of the sea slugs were fed algae treated the same way, except that 5 mM NaF was added after 10 min of PSII light pre-illumination to inhibit the dephosphorylation-dependent state 2 to state 1 transition (see Fig. 4e for a schematic of the experiment). However, kleptoplasts in sea slugs from both feeding groups were in state 1 (Fig. 4f), indicating that state 2 to 1 reversion occurred almost immediately upon feeding and is independent of phosphatase activity.

## Chloroplast shape changes coincide with a shift from state 2 towards state 1

Continuously red light illuminated *A. acetabulum* algae in state 2 were subjected to altered salinity environments to test if osmotic changes can force the chloroplasts to state 1 (Fig. 5). In regular artificial sea water (ASW) with a salinity of 35 PPT (parts per thousand) the 20 min red PSII light treatment resulted in very strong state 2 (Fig. 5a). However, when the algae were first illuminated with red light for 15 min in 35 PPT ASW and then placed to ASW with higher (40 and 45 PPT; Fig. 5b) or lower salinity (30 and 25 PPT; Fig. 5c) for an additional 5 min, still under red PSII light, the light acclimation state switched to an intermediate state between state 1 and 2. To ensure that the salinity treatments were not destroying the chloroplasts, photosynthetic electron transfer (relative electron transport rate - rETR) was also estimated from the algae by measuring rapid light response curves with a PAM fluorometer. The highest salinity, 45 PPT, and the second lowest salinity of 30 PPT had only minor effects on the rETR of *A. acetabulum*, but the lowest

salinity treatment, 25 PPT, had a detrimental effect on the photosynthesis of the algae (Fig. 5d–f). The slightly increased salinity, 40 PPT, seemed to stimulate photosynthesis in the algae (Fig. 5e). When the sea slug *E. timida* was maintained at 35 PPT ASW, rETR was also greatly stimulated in comparison to the algae they had been feeding on (Supplementary Fig. 8).

To measure the effects of salinity changes on chloroplast shape, chlorophyll *a* auto-fluorescence in live *A. acetabulum* was imaged using confocal microscopy (Fig. 6). The shapes of the chloroplasts in algae kept at the regular 35 PPT salinity were highly variable, but many of the chloroplasts were elongate, containing two ovoid parts connected with a constricted section in the middle. A slight overall narrowing of the chloroplasts was noticeable when the algae were exposed to an increased salinity of 40 PPT, but the shapes were similarly diverse as in the 35 PPT treatment. The most noticeable shape change took place in the 30 PPT lowered salinity treatment, where the chloroplasts appeared rounder, although still noticeably ovoid (Fig. 6a). However, when the circularity of the chloroplasts was analyzed (defined as $(4\pi \times$ Chloroplast Area)/Chloroplast Perimeter²; values between zero and one, one being a perfect circle), only the difference between the 40 PPT and 30 PPT treatments was statistically significant (Fig. 6c). Compared to the alga itself, *A. acetabulum*-derived kleptoplasts in *E. timida* and *E. crispata* were remarkably uniform in shape (Fig. 6b) and seemingly smaller in size compared to *A. acetabulum* chloroplasts (Supplementary Fig. 9). Both sea slug species had nearly perfectly circular kleptoplasts, significantly differing from both 35 PPT and 40 PPT chloroplasts in *A. acetabulum*, whereas kleptoplasts of neither sea slug showed significant differences in circularity in comparison to the swollen chloroplasts of the alga in 30 PPT salinity (Fig. 6c).

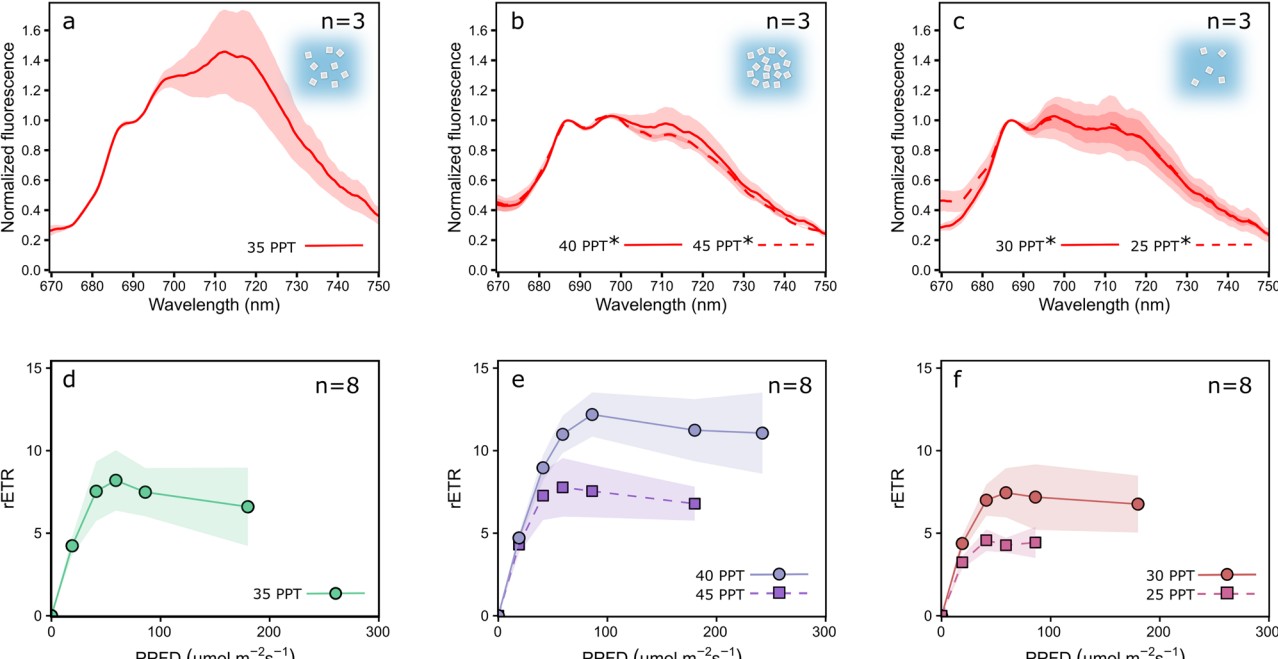

**Fig. 5 | The effect of salinity changes on state transitions and photosynthetic electron transfer in *Acetabularia acetabulum*. a** 77 K fluorescence of *A. acetabulum* samples after exposure to 20 min of red PSII light (PPFD 50 μmol m$^{-2}$ s$^{-1}$) in ASW with a salinity of 35 PPT (control). 77 K fluorescence from *A. acetabulum* treated identically to the samples used in (**a**), except for a switch to higher (40 or 45 PPT, **b**) or lower salinity (30 or 25 PPT, **c**) for the final 5 min of the 20 min PSII light treatment. The asterisks next to the salinity descriptions in (**b**, **c**) mark significant differences in normalized PSI fluorescence between the indicated salinity treatments and the 35 PPT treatment in (**a**) (one-way ANOVA followed by post-hoc Tukey's test, *$p$ value < 0.05). **d–f** Relative electron transfer rate (rETR) of *A. acetabulum* measured with a PAM fluorometer after a 5 min treatment in growth conditions in the salinities described in the panel legends. The illumination at each PPFD lasted 60 s before determining rETR with a saturating light pulse, and rETR values are only shown down to the light intensity where the rETR was above zero in all replicates. The number of biological replicates (*n*) is shown in the panels. The shaded areas around the curves show standard deviation.

## Functional kleptoplasts are near-perfect spheres

The possible universality of the chloroplast shape changes from an irregular ovoid inside the algae to a perfect circle, or sphere, inside kleptoplastic sea slugs was tested by inspecting other ulvophyte algae and sacoglossan sea slugs with a confocal microscope (Fig. 7). The chloroplasts of the Bryopsidales alga *B. plumosa* were thin, platelet-like ovoid shapes inside the alga, but near-perfect circles inside the sea slug *E. crispata* (Fig. 7a, b). Furthermore, the kleptoplasts appeared smaller than the chloroplasts inside the alga, similarly to what was observed above with *A. acetabulum*, but the difference in size between the Bryopsidales chloroplasts inside these two hosts was a lot more notable than in any other tested alga-sea slug combinations (Fig. 7c; see also Supplementary Fig. 9 for the chloroplast sizes from other algae and sea slugs). Finally, the chloroplasts from another Bryopsidales alga, *C. tomentosum*, were inspected in the alga itself and inside the functionally kleptoplastic sea slug *E. viridis*, but also inside another sacoglossan sea slug, *P. dendritica*, that is not capable of functional kleptoplasty. The chloroplasts of the alga *C. tomentosum* resembled those of *B. plumosa* but appeared smaller in size. Once again, the chloroplasts transformed to almost perfect circles inside the sea slug *E. viridis*, whereas their shape was less circular and more varied in the non-functional kleptoplast retention species *P. dendritica*, although still significantly more circular than in the alga (Fig. 7d, e).

## Discussion

Losing some of the commonly available photoprotection mechanisms does not seem like an advisable evolutionary strategy for any organism trying to utilize sunlight for energy, especially when inhabiting marine intertidal regions where light conditions often fluctuate excessively. Nevertheless, in our present work, we demonstrate that one entire order of green macroalgae, Bryopsidales, as well as kleptoplastic sacoglossan sea slugs eating any ulvophyte algae, seem to keep on doing exactly that. As summarized in

Fig. 8a, we show here that Bryopsidales have lost state transitions, in addition to the previously shown loss of the qE component of NPQ[28,29]. Interestingly, the shell-bearing sacoglossans (Oxynoacea), which have retained ancestral characteristics, including the absence of functional kleptoplasty, feed exclusively on Bryopsidales algae such as *Halimeda* or *Caulerpa*. The diversification of diets to include also true ulvophytes seems to have taken place during the evolution of the non-shelled sacoglossans (Plakobranchacea) that include all functionally kleptoplastic species[16,42]. Perhaps the lack of some of the complex photoprotection and acclimation processes in the chloroplasts of Bryopsidales made them particularly suitable for the initial incorporation of chloroplasts by the first kleptoplastic sea slugs that did not possess the fine-tuned mechanisms required for the maintenance of the more dynamic photoprotection in true ulvophytes.

## Bryopsidales have lost state transitions during evolution

Surprisingly, we did not see any changes in the 77 K PSI/PSII fluorescence of any Bryopsidales algae in response to the red and far-red light treatments (Fig. 2). The wavelength preferences of PSII and PSI are known to be largely controlled by the chlorophyll *a* and *b* ratios of their antennae[39], as PSII-LHCII of plants, for instance, contains more chlorophyll *b* than PSI-LHCI. The differences in PSII and PSI absorption profiles might be more nuanced in green algae like *C. reinhardtii*, but the 660 nm red and far-red (≥700 nm) lights used here have been shown to be consistent in preferentially exciting PSII and PSI in multiple green algal species[43,44]. While the LHCII antenna of Bryopsidales (also called siphonaxanthin–chlorophyll *a/b*-binding protein; SCP) differs significantly from that of other green algae, similar, if not larger, differences in the PSII and PSI absorption profiles are expected to exist in Bryopsidales due to the extra chlorophyll *b* in the LHCII of Bryopsidales compared to other green algae[45,46], suggesting that the lack of observable state transitions did not derive from the used lights not favoring PSII and PSI in Bryopsidales.

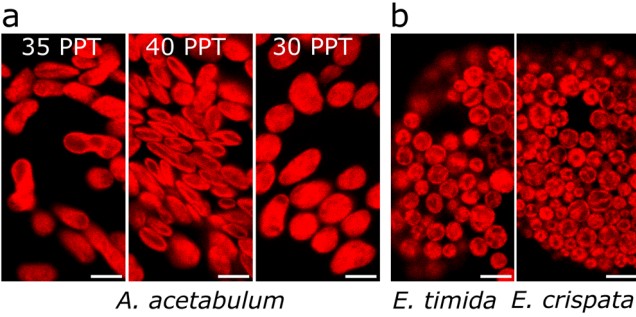

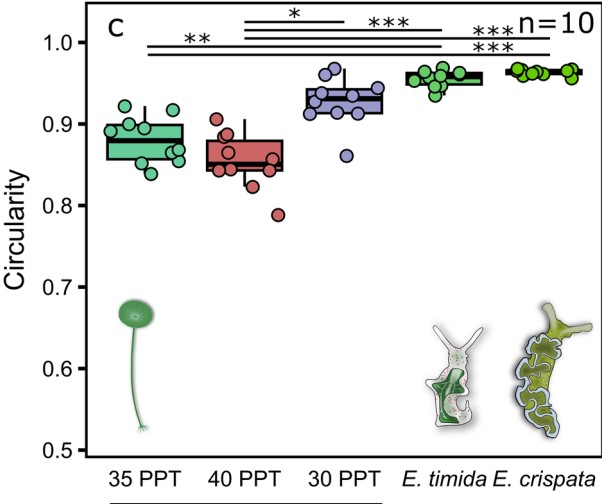

**Fig. 6 | The circularity of *Acetabularia acetabulum* chloroplasts before and after osmotic shocks and kleptoplastic incorporation by photosynthetic sea slugs, imaged with confocal microscopy using chlorophyll *a* fluorescence.** Exemplary images from **a** *A. acetabulum* in ASW with different salinities, 35 PPT being the growth condition, or from **b** sea slugs *Elysia timida* and *Elysia crispata* with *A. acetabulum*-derived kleptoplasts. The scale bars are 5 μm in all representative images in (**a**, **b**), and brightness and contrast adjustments were applied to the entire images for visual purposes. **c** The circularity of the chloroplasts in the different salinity treatments of *A. acetabulum* and the sea slugs, calculated as $(4\pi \times$ Chloroplast Area)/ Chloroplast Perimeter$^2$, where the value for individual chloroplasts varies between 0 (not a circle) and 1 (a perfect circle). Each individual datapoint shows the median chloroplast circularity from a single biological replicate (calculated from hundreds to thousands of identified chloroplasts), whereas the box plots show the medians and interquartile ranges from all replicates. The number of biological replicates (*n*) is shown in (**c**). The whiskers indicate non-outlier maxima and minima. Asterisks mark significant differences between the indicated treatment groups (Kruskal–Wallis, followed by post-hoc Dunn's test, $*p$ value < 0.05, ** < 0.01, *** < 0.001).

The qL parameter is admittedly not a fully reliable probe of the PQ pool reduction during monochromatic light treatments[39]. However, the big differences in qL during red and far-red light treatments in both Bryopsidales and true ulvophytes (Fig. 3a) do suggest that the light treatments caused reduction and oxidation of the PQ pool in most of the algae, but state transitions only occurred in true ulvophytes (Fig. 2). The near ubiquitous existence of state transitions in green algae and plants would suggest that the Bryopsidales have lost this capacity at some point during their evolution, possibly at the same time as the loss of qE[29].

Interestingly, our additional experiments with the true ulvophyte alga *Chaetomorpha* sp. suggest a loss of qE also within some true ulvophyte algae; the thylakoid proton gradient uncoupler nigericin had no effect on NPQ in this alga, while the xanthophyll cycle inhibitor DTT led to a diminished NPQ induction during high light (Supplementary Fig. 6). The reliance on a

functional xanthophyll cycle for NPQ agrees with a recent study in *Chaetomorpha* sp., where conversion of violaxanthin to zeaxanthin was observed in high light by Morelli et al.[24]. Morelli et al.[24] did not suggest a lack of qE based on NPQ relaxation kinetics in *Chaetomorpha* sp., but no uncouplers were used to probe the effects of the proton gradient. It should be noted that nigericin would be expected to also inhibit the conversion of violaxanthin to zeaxanthin if the violaxanthin de-epoxidase of *Chaetomorpha* sp. was similarly dependent on the acidification of the thylakoid lumen as in plants. However, the enzyme repertoire for violaxanthin de-epoxidation in green algae is diverse[47], and some of the enzymes, like the stromal lycopene cyclase homolog in *C. reinhardtii*, are not dependent on thylakoid lumen acidification[48]. Based on our results, *Chaetomorpha* sp. is an interesting subject for future studies discovering the true diversity of the xanthophyll cycle and its link to NPQ in green algae.

The loss of state transitions in Bryopsidales might be related to the structure of their antenna proteins, possibly rendering state transitions infeasible. Indeed, the Bryopsidales PSI-LHCI structure is missing the PsaO protein at the supposed binding location of LHCII to PSI[49,50]. The fitness advantages gained with this loss in light acclimation capacity remain, however, obscure. Clearly this has not hindered the success of Bryopsidales too much, as some of them, like *Caulerpa taxifolia*, are considered highly invasive species[51].

One example of a green alga that does not have state transitions is the Antarctic *Chlamydomonas* sp. UWO241, adapted to perform photosynthesis in a cold, dim light and high salinity environment[52,53]. This alga is particularly interesting due to its almost identical 77 K fluorescence spectrum with the Bryopsidales alga *B. plumosa*, with both lacking the PSI-associated fluorescence emission at 710–730 nm (Fig. 3d)[54]. Furthermore, heat stress has been shown to lead to similar changes in the 77 K fluorescence spectrum in UWO241 as those we noticed in *B. plumosa*-derived kleptoplasts of *E. crispata* (Fig. 3e)[52]. This certainly draws parallels between UWO241 and *B. plumosa*, since in our facilities, *B. plumosa* is grown at 20 °C, whereas *E. crispata* is fed and grown at 25 °C. It is tempting to speculate that the suggested adaptations related to the loss of state transitions in UWO241, like constitutive cyclic electron flow around PSI and reliance on energy spill-over between PSII and PSI[53,55,56], also translate to Bryopsidales. Such phenomena should be investigated, but the different habitats of UWO241 and the intertidal Bryopsidales may, however, suggest a very different environmental pressure in selecting for the state transitions deficiency between the two algae.

## State transitions are lost during kleptoplast incorporation

Differences in photosynthesis and photoprotection are known to exist between kleptoplastic sea slugs and their algal prey. For instance, photosynthetic electron transfer tends to be faster in *E. viridis* than in its Bryopsidales kleptoplast donor alga *C. tomentosum*[57], although the differences are sometimes small[24]. Furthermore, the PQ pool has been suggested to remain more oxidized in *E. timida* compared to *A. acetabulum*, which might help limit ROS production[30]. Our new qL data suggest that maintaining the PQ pool relatively oxidized is a general tendency in sea slug kleptoplasts compared to their prey algae, except for possibly *E. crispata* and *B. plumosa* (Fig. 3a, b, e, f). However, previous research has usually observed more NPQ in the sea slugs than in their prey algae; the opposite of what we noticed in our new data showing high levels of NPQ in *A. acetabulum*, but not in *E. timida* or *E. crispata* (Fig. 3g, h)[21,22,30]. To solve this discrepancy, we did an additional experiment comparing NPQ induction of *A. acetabulum* and *E. timida* under different wavelengths of light. Indeed, NPQ might be enhanced in the sea slugs in white and blue light, but not in green or red light (Supplementary Fig. 5). Some of these differences likely relate to the different optical properties of the specimens[58], but in the light of our new state transition data it is highly plausible that the algae and the sea slugs respond to light very differently also at the level of light-harvesting complex functionality.

Certain unicellular kleptoplastic dinoflagellates may represent a precedent for the loss of state transitions in kleptoplastic organisms. Although it

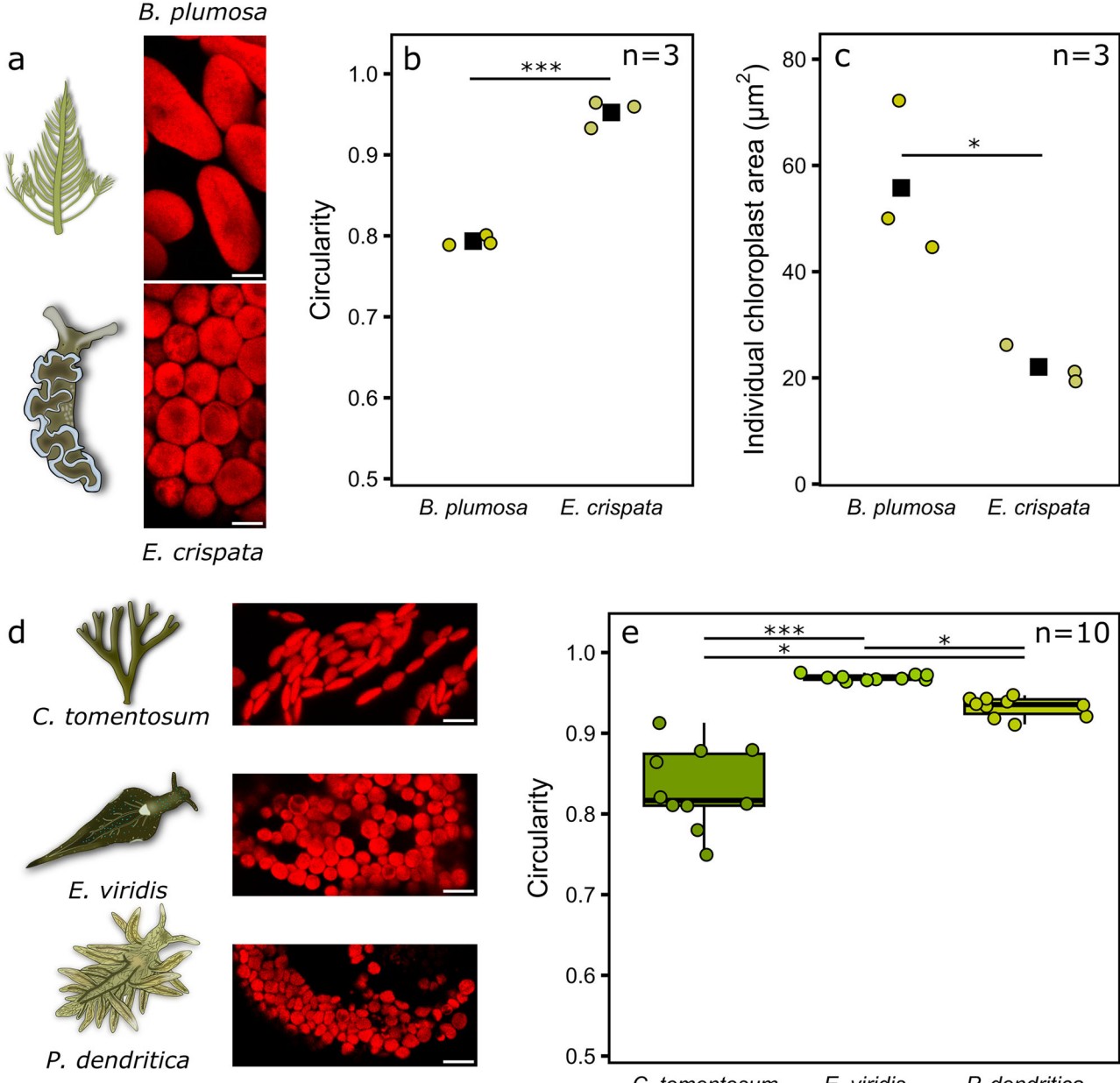

**Fig. 7 | The shapes and sizes of Bryopsidales chloroplasts in the algae and inside the kleptoplastic sea slugs. a** Chloroplast imaging with a confocal microscope using chlorophyll *a* fluorescence from the alga *Bryopsis plumosa* itself (top), and after kleptoplastic incorporation into *Elysia crispata* cells (bottom). **b** The circularity of the chloroplasts in *B. plumosa* and in the sea slug *E. crispata*. **c** An estimation of the individual chloroplast area for the chloroplasts in *B. plumosa* and *E. crispata*. The circles and black squares show the individual data points (the median chloroplast circularity or area from a single biological replicate) and their mean values, respectively, in (**b**, **c**). **d** Exemplary confocal microscope images of *Codium tomentosum* chloroplasts inside the alga *C. tomentosum* (top), the kleptoplastic sea slug *Elysia viridis* (middle), and the sea slug *Placida dendritica* that is not capable of functional kleptoplasty (bottom). **e** The circularity of the chloroplasts in *C.*

*tomentosum* and in the two sea slug species. The circularity was calculated as $(4\pi \times \text{Chloroplast Area})/\text{Chloroplast Perimeter}^2$, where the value for an individual chloroplast varies between 0 (not a circle) and 1 (a perfect circle). In (**e**), each individual datapoint shows the median chloroplast circularity or area from a single biological replicate (calculated from hundreds to thousands of identified chloroplasts). The box plots show the medians and interquartile ranges from all biological replicates. The number of biological replicates (*n*) is shown in the panels. The whiskers indicate non-outlier maxima and minima. The asterisks mark significant differences between the indicated groups, determined by Student's *t*-test for (**b**, **c**) and Kruskal–Wallis, followed by Dunn's test for (**e**) (*$p$ value $< 0.05$, *** $< 0.001$). The scale bars are 5 μm in all representative images in (**a**, **d**), and brightness and contrast adjustments were applied to the entire images for visual purposes.

is debatable whether state transitions occur in algae with complex chloroplasts of secondary endosymbiotic origin[59], Stamatakis et al.[60] reported spectroscopic data in haptophyte chloroplasts inside a kleptoplastic dinoflagellate that were interpreted as a loss of some form of state transitions. On the other hand, state transitions in the true ulvophyte *A. acetabulum* are strong and dynamic, dependent on phosphorylation (Fig. 4a–d), and their loss in kleptoplastic sea slugs is very clear (Fig. 2k, l). Furthermore, our

feeding experiment with bleached *E. timida* individuals in the presence of the phosphatase inhibitor NaF shows that kleptoplast incorporation immediately enforces state 1, regardless of the original light state in *A. acetabulum* chloroplasts (Fig. 4e, f). Even if NaF was not reaching the kleptoplasts in the sea slugs, the fact that the NaF and red light-treated *A. acetabulum* chloroplasts were locked in state 2, which was irreversible even under far-red light (Fig. 4d), shows that the sea slugs do alter the light

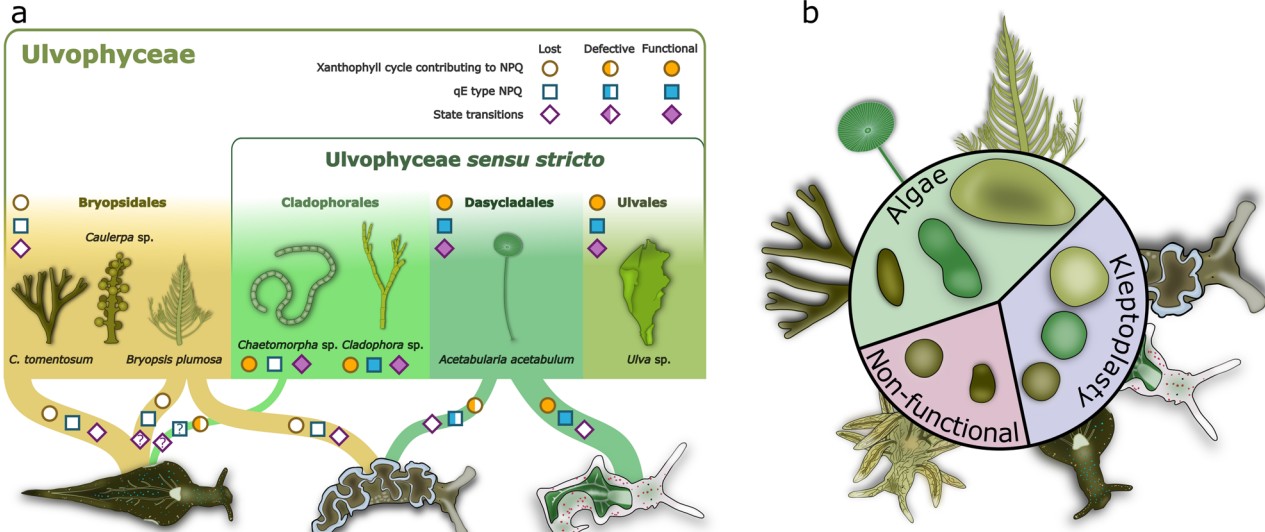

**Fig. 8 | Summary of the major differences in photoprotection, light acclimation, and chloroplast shapes between Bryopsidales, other ulvophytes, and both functional and non-functional kleptoplastic sea slugs hosting the kleptoplasts derived from ulvophyte macroalgae. a** The existence and functionality of the xanthophyll cycle contributing to non-photochemical quenching (NPQ), the energy-dependent qE component of NPQ, and state transitions in Ulvophyceae and the kleptoplastic sea slugs feeding on different ulvophytes. The differently colored circles and squares describe the different levels of the xanthophyll cycle and qE, respectively, in different algal groups/sea slugs, as described in Christa et al.[29] and Morelli et al.[24,25], and here for *Chaetomorpha* sp. The occurrence of state transitions is indicated by the differently colored diamonds and relies solely on the data from the present study. Question marks indicate untested properties. The thickness of a node connecting an alga and a sea slug approximates the strength of the prey-predator relationship of the species in our laboratory conditions. The division of Ulvophyceae macroalgae to the core set of ulvophytes (Ulvophyceae *sensu stricto*) and to Bryopsidales (see Fig. 1) is based on the classification by Hou et al.[18]. **b** The chloroplasts inside the ulvophyte macroalgae show highly varied shapes, whereas the kleptoplasts inside the sea slugs are nearly perfect spheres. Sea slugs that are incapable of functional kleptoplasty have less circular kleptoplasts than slugs with functional kleptoplasty.

acclimation state of the chloroplasts during incorporation. This suggests that the alterations to the kleptoplasts go beyond the control of the PQ pool redox state and the phosphorylation/dephosphorylation of the mobile LHCII antenna. It is unclear whether the loss of state transitions in kleptoplastic dinoflagellates and sea slugs has anything in common, but it is tempting to speculate on the universalness of the phenomenon in kleptoplasty.

**Structural changes underpin the loss of state transitions in kleptoplasts**

Salinity changes have many effects on green algae[61], but two relatively common responses are the enforcement of either light acclimation state 1 or 2[62–64], and alterations in the chloroplast ultrastructure[53,63,65]. Unlike in the green microalga *Dunaliella salina*, where only lowered salinity has been shown to induce state 1[62], in *A. acetabulum* both low and high salinity treatments caused a strong transition towards state 1 (Fig. 5). Both high and low salinity treatments of *A. acetabulum* also caused structural changes to the chloroplasts (shrinkage and swelling, respectively; Fig. 6a, c, Supplementary Fig. 9), making the altered ultrastructure of chloroplasts a plausible mechanism for the removal of state transitions as a side effect; an osmotic influx of water might also explain the near perfectly circular shapes of kleptoplasts (Fig. 6b, c). Most marine invertebrates are osmoconformers, and photosynthetic sea slugs are seemingly no exception to this[66], making it admittedly unlikely that the sea slugs would alter the osmotic environment of their kleptoplasts to the same extent as our salinity treatments. On the other hand, macroalgae with rigid cell walls may be able to maintain turgor pressure by regulating ion movements across the plasma membrane and/or by synthesizing osmoregulators[67–69]. Thus, chloroplasts removed from algal cells during feeding may experience a hypoosmotic shock, possibly explaining their spherical shape inside the sea slugs (Figs. 6 and 7). However, kleptoplasts tend to be rather smaller in size than the original chloroplasts (Fig. 7; Supplementary Fig. 9), suggesting that an osmotic shock is at least not the whole answer.

An alternative explanation is that the suctorial feeding style of the sea slugs exposes the chloroplasts to strong mechanical stress, perhaps concomitantly with mild osmotic changes, that selects for a specific shape and size. This could explain why the kleptoplasts of *E. timida* and *E. crispata* seem to have characteristics of both high and low salinity *A. acetabulum* chloroplasts; the kleptoplasts are very spherical and look slightly swollen, as expected for low salinity (Fig. 6), but they have also seemingly shrunk in size (Supplementary Fig. 9), and the photosynthetic electron transfer of *E. timida* kleptoplasts is stimulated in a similar manner as in *A. acetabulum* in high salinity (Fig. 5e, Supplementary Fig. 8). Based on the few available microscope images of chloroplasts still in the gut lumen of sacoglossan sea slugs prior to incorporation as kleptoplasts[70,71], it seems likely that the shape change to a perfect sphere happens already in the gut lumen, which agrees with the witnessed immediate transition to state 1 of the sea slug kleptoplasts (Fig. 4f).

**Benefits of compact and spherical kleptoplasts?**

The striking chloroplast/kleptoplast shape change between algae and sea slugs has been previously reported, but its implications for kleptoplast photosynthesis and longevity have not been discussed[72–74]. Here, we show that the change in circularity takes place in all tested photosynthetic sea slugs and seems to be a fundamental property of kleptoplasty in these animals (Figs. 6 and 7). It seems likely that the kleptoplasts are also smaller than the chloroplasts, especially in *E. crispata* fed with *B. plumosa* (Fig. 7a, c, Supplementary Fig. 9), but final conclusions about their volumes will require 3D imaging. McLean[70] showed that the chloroplasts are also highly circular in the gut lumen of *P. dendritica*, a sea slug species incapable of functional kleptoplasty, but become unstructured almost immediately after phagosomal encapsulation due to digestive degradation[75]. Similar findings also exist from *Thuridilla hopei*, another sacoglossan non-functional kleptoplast retention species[76,77]. Our data show that the chloroplasts inside *P. dendritica* are indeed more circular than inside *C. tomentosum*, but less circular than *E. viridis* kleptoplasts derived from the same source (Fig. 7d, e), corroborating

the earlier findings. This suggests that the capacity for sphericity enforcement is inherent in Sacoglossa, but only functionally kleptoplastic species may make full use of it (Fig. 8b).

Unlike ulvophyte macroalgae, photosynthetic sea slugs are not sessile organisms and can move away from unfavorable light conditions[78]. Losing the photoprotection provided by the slowly inducible state transitions because of structural changes might therefore not be detrimental to sea slug kleptoplasts, especially since their loss seems to come with the benefit of a more oxidized PQ pool (Fig. 3), more efficient electron transfer (Fig. 5, Supplementary Fig. 8)[57], and enhanced NPQ in blue light that dominates in marine environments (Supplementary Fig. 5). Indeed, kleptoplasty in sacoglossans seems to have originally evolved in species feeding only on Bryopsidales algae that lack state transitions[16,42], which suggests that relying on state transitions for photoprotection would only introduce an unnecessary level of complexity for kleptoplast maintenance. The full complement of alterations the structural changes impose on photosynthetic protein complexes, their interactions and repair processes remain to be investigated, but one advantage gained from a spherical shape could simply be enhanced structural integrity that helps the kleptoplasts stay intact as long as possible.

Whether the forced sphericity of chloroplasts is an absolute requirement for functional kleptoplasty in sea slugs remains an open question, but it does take place in all studied kleptoplastic sea slug-algae associations. We propose that it is likely a fundamental property supporting long-term retention of kleptoplasts in photosynthetic sea slugs.

## Methods

### Organisms and culture conditions

The main species experimented on in this study were the sea slugs *E. timida* and *E. crispata*, and their feedstock algae *A. acetabulum* and *B. plumosa*. The sea slug *E. timida* was originally collected from the Mediterranean (Elba, Italy) and has been routinely cultured for generations in our laboratory conditions, essentially as previously described[30]. The *E. crispata* population originates from Florida, USA, and was purchased from TMC Iberia (Lisbon, Portugal), after which new generations have been routinely cultured in our laboratory as described earlier[25,27]. The Mediterranean alga *A. acetabulum* (strain DI1) was originally isolated by Diedrik Menzel and was maintained as described earlier[30]. *B. plumosa* was grown as described by Cartaxana et al.[27]. The salinity of all ASW (Red Sea salt, Red Sea Europe, Verneuil d'Avre et d'Iton, France) used for growing the sea slugs and algae was 35 PPT unless otherwise mentioned. As a default, *E. crispata* was fed with *B. plumosa*, and *E. timida* with *A. acetabulum*. Some of the experiments required the use of *E. crispata* individuals containing chloroplasts derived from *A. acetabulum*. In these cases, the selected *E. crispata* individuals were allowed to feed exclusively on *A. acetabulum* for a minimum of 10 days prior to the experiments, as this has been shown to be a sufficient period for an almost total switch in the chloroplast content of this sea slug species[24,27].

Other sea slug and algae species used were collected from the wild and only transiently cultured in the laboratory for the experiments. In these cases, the specimens were allowed to acclimate to the laboratory conditions for a minimum of 1 week before the experiments. The details of all the sea slug and algae species used in the study, their original locality, and their growth conditions are summarized in Supplementary Tables 1 and 2, respectively.

### Bleaching of *E. timida*

A subset of 100 adult *E. timida* individuals was selected and put under high light (PPFD 1500 µmol m$^{-2}$ s$^{-1}$; 50 W cold white LED flood light, ExtraStar Electrical Ltd, Salford, England) in a 12/12 h day/night cycle for 11 days without any algae to feed on. The 500 mL glass flasks containing the sea slugs were sealed with parafilm and constantly sparged with air during the 11-day period. An increase in salinity caused by the small amount of ASW evaporation from the flasks was counteracted every 2 days by adding an equivalent amount of distilled H$_2$O, and the flasks and ASW were completely changed every 4 days. After 11 days, the sea slugs were visually

inspected and 54 palest individuals were selected (Supplementary Fig. 7) for a re-feeding experiment (see below).

### Induction of state transitions

Two custom built LED systems (Shenzhen LED Color Co., LTD, Shenzhen, China) were used to induce state transitions in the sea slugs and the algae: a 660 nm red light that preferentially excites PSII (PSII light), and a 740 nm far-red light for preferential PSI excitation (PSI light) (see Fig. 2a for the spectra). The absolute irradiance of both types of light was measured using a calibrated FLAME-T-VIS-NIR miniature spectrometer equipped with a cosine corrected QP400-1-VIS-NIR light guide (Ocean Insight, Orlando, FL, USA), converted to PFD and set to 50 µmol photons m$^{-2}$ s$^{-1}$ unless otherwise mentioned. The main protocol for the induction of state transitions consisted of placing the samples in the dark for 15 min, and then exposing them for 15 min to either PSII or PSI light, but different combinations of darkness, PSII, and PSI treatments were also applied as described in the text. All light treatments were done at ambient room temperature. Depending on the size of the sea slug species, different numbers of individuals were placed in a well of a transparent 6- or 24-well plate filled with ASW to ensure sufficient biomass for downstream analyses of the samples. With *E. crispata*, a large sea slug with an approximate adult body length of >20 mm, only one animal was placed in a well, whereas three to five individuals of the smaller sea slugs *E. timida* (adult body length ≤ 10 mm) and *E. viridis* (≤15 mm) were placed in a well. After the desired light treatment, all sea slugs from a single well were immediately frozen and ground to a fine, homogenous powder in liquid nitrogen and collected into a pre-cooled cryotube that was thereafter never allowed to thaw and kept as close to 77 K temperature as possible. The pulverized animals originating from a single well were treated as a single biological replicate in the downstream analyses of state transitions. The protocol was essentially the same for the algae, although special consideration had to be paid to not cause substantial self-shading of the algal filaments.

In some of the experiments, the samples were exposed to 5 mM concentration of the phosphatase inhibitor NaF (CAS#: 7681-49-4; Merck, Darmstadt, Germany) that prevents the dephosphorylation of LHCII and therefore also the reversal of light acclimation from state 2 to state 1[79]. In these cases, the *A. acetabulum* algae were always first treated with 15 min of darkness and 10 min of red PSII light prior to pipetting NaF from a 1 M stock solution (in H$_2$O) to the ASW and gently mixing by swirling the experimental vial. The algae were then exposed to an additional 5 min of red light before a switch to a 15 min treatment with far-red light before sampling. In the feeding experiment with bleached *E. timida*, the algae were also kept under the red light for 5 min after the addition of NaF, after which the sea slugs were added to the same vessel to feed on the algae, always under the red light and exposed to 5 mM NaF. The animals were constantly inspected for color changes from bleached to green, and every time 3–4 individuals had turned sufficiently green to provide enough photosynthetic material for the 77 K fluorescence measurements, they were picked out and ground to a powder in liquid nitrogen. The first sampling point was ~20 min after the feeding had commenced, whereas the last sampling point was after 120 min of feeding. The other samplings were done at infrequent intervals.

The effect of salinity on state transitions in *A. acetabulum* was tested by first placing dark acclimated algae under the red PSII light for 15 min in ASW with a salinity of 35 PPT (same as their growth conditions), after which the algae were transferred to ASW with either 25, 30, 40, or 45 PPT salinity and maintained under the PSII light for an additional 5 min. The samples were then frozen and ground in liquid nitrogen as usual. The salinities were measured with a HI96822 digital refractometer (Hanna Instruments, Woonsocket, RI, USA).

### Chlorophyll fluorescence spectra at 77 K

Because the chlorophyll concentration of the pulverized samples from the state transition induction experiments was unknown, all samples had to be serially diluted to ensure as little self-absorption of chlorophyll fluorescence as possible during the subsequent fluorescence measurements at liquid

nitrogen (77 K) temperature. For this, the samples were diluted with optically neutral quartz sand ($SiO_2$; CAS#: 14808-60-7, Merck). For most samples the dilution series consisted of 100% (pure sample powder taken straight from the grinding), 50% (a rough 50/50 mixture of sample and $SiO_2$; v/v) and 25% (a rough 25/75 mixture of sample and $SiO_2$) concentrations, but heavily concentrated samples had to be diluted even further. The total volume of the sample and $SiO_2$ varied from ~50 to 200 µL. The $SiO_2$ was pre-cooled in liquid nitrogen before adding it to the sample powder inside a 2 mL cryotube, and all samples were vortexed in short bursts before the fluorescence measurements to thoroughly mix the sample powder with the $SiO_2$.

One screw-on cap of a 2 mL cryotube was punctured, and the probe end of a bifurcated R600-7-VIS-125F light guide (Ocean Insight) was fitted to the hole so that the cap surrounded the probe end at a desired height. The samples in the cryotubes were then fastened to the cap on the probe so that the sample powder at the bottom of the tube was in contact with the probe. The sample tube and the probe were then submerged in liquid nitrogen inside a box that was lined and covered with black cloth. The blue excitation light for all measurements was obtained by filtering the light from a 50 W white LED flood light (ExtraStar Electrical Ltd) with 500 nm low-pass (500FL07-50S; Andover Corporation, Salem, NH, USA) and 450 nm band-pass (450FS20-12.5, Andover Corporation) filters (see Supplementary Fig. 3 for the excitation light spectrum). The excitation light was guided to the sample through the probe by placing the other end of the R600-7-VIS-125F light guide perpendicular to the 450 nm band-pass filter. The third end of the light guide was connected to a FLAME-T-VIS-NIR spectrometer equipped with a 100 µm aperture assembly-slit. Chlorophyll fluorescence was recorded using the OceanView software with the following settings: integration time 10 s, boxcar width 3, scans to average 1, and the non-linearity correction was on. Before recording, the fluorescence signal was allowed to stabilize for ~2 min or until visible changes in the shape of the spectrum were no longer noticeable.

The data from the dilution series of a single biological replicate were inspected to estimate the effect of self-shading on chlorophyll fluorescence. In an ideal situation, the data from the 50% and 25% sample concentrations showed the same shape of the chlorophyll fluorescence spectrum, validating the use of the less noisy 50% data. However, sometimes the spectrum did still show slight differences in shape between 50% and 25% sample concentrations, especially in the 710–730 nm region, indicating self-absorption of chlorophyll fluorescence and making the 50% sample concentration data unusable. In such cases, the most dilute sample concentration that still exhibited noticeable chlorophyll fluorescence was selected for downstream analyses. The fluorescence data were then processed in MS Excel v.16.0 (MS Office 365; Microsoft Corporation, Redmond, WA, USA) by applying smoothing with a 4 nm running average, setting the zero-level signal by subtracting the signal at 800 nm from the entire spectrum, and normalizing the data to the PSII fluorescence peak between 680 and 690 nm.

## Photosynthesis measurements

Different parameters related to photosynthesis were obtained using routine saturating pulse analyses with PAM fluorometers that record variable chlorophyll *a* fluorescence from photosynthetic material. Briefly, the core parameters of the saturating light pulse analyses were the PSII activity parameters $F_V/F_M$, i.e., $(F_M - F_0)/F_M$, where $F_M$ and $F_0$ are the maximum and minimum fluorescence from dark acclimated samples, and $(F_M' - F)/F_M'$, where $F_M'$ and $F$ are the maximum and transient fluorescence during continuous illumination. NPQ was calculated as $F_M/F_M' - 1$. The fraction of PSII reaction centers in the open state was estimated using the qL parameter[40], defined as $((F_M' - F)/(F_M' - F_0')) \times (F_0'/F)$, where $F_0'$ is a theoretical estimation of the true minimum fluorescence of a light acclimated sample, calculated as described by Oxborough and Baker[80].

Variable chlorophyll *a* fluorescence during illumination with the external PSII and PSI lights was measured using Junior-PAM (Heinz Walz GmbH, Effeltrich, Germany) from samples that were kept in low light (PPFD ~ 10 µmol m$^{-2}$ s$^{-1}$) for a minimum of 15 min prior to the

measurements. Rapid light curves were measured from samples taken from their growth conditions without dark acclimation using Imaging-PAM (Mini version; Heinz Walz GmbH). The duration of the light steps for each actinic light intensity was 60 s, after which a saturating light pulse was fired to estimate rETR of PSII, calculated as $0.42 \times ((F_M' - F)/F_M') \times$ PPFD, where 0.42 is a plant-based estimation of the fraction of incident photons absorbed by PSII. Rapid light curves were also measured from *A. acetabulum* exposed to either 25, 30, 40, or 45 PPT salinity ASW. Here, the algae were put to the salinity treatment in their growth conditions for 5 min and then exposed to the rapid light curve protocol. The algae were kept in ASW with the desired salinity until the end of the protocol. The fluorescence kinetics during and after a high light (PPFD 500 µmol m$^{-2}$ s$^{-1}$) treatment were measured from *Chaetomorpha* sp. using Imaging-PAM in the absence and presence of the uncoupler nigericin (60 µM; CAS#: 28643-80-3, Merck) and the reducing agent 1,4-dithiothreitol (DTT, 10 mM; CAS#: 3483-12-3, Roche Diagnostics GmbH, Basel, Switzerland). The algae were incubated in ASW in the absence or presence of either one of the reagents for 30 min in the dark prior to the high-light treatment.

The measuring beam lights of both Junior-PAM and Imaging-PAM were blue, as well as the actinic light of Imaging-PAM. The immobilization of the sea slugs during the fluorescence measurements was achieved by placing them on a black cloth on top of tissue paper. The tissue paper sucked out most of the moisture through the cloth, rendering the animals sufficiently immobile for the duration of the measurements.

## Confocal microscopy and image analysis

The algae samples were prepared for microscopy by placing one to three live algal filaments or pieces on a regular rectangular microscope glass slide, adding a drop of ASW on the samples, and then placing a cover glass on top. When the effect of salinity on chloroplast characteristics was inspected in *A. acetabulum*, the algae were put in ASW with the desired salinity for at least 10 min before the filamentous cells were prepared on the slides. Here, the salinity of the ASW that was added to the slide was always the same as in the salinity treatment in question. *E. timida* and *E. crispata* individuals were taken for imaging 1–3 days after supplying them with their food. *E. viridis* and *P. dendritica* were continuously fed with *C. tomentosum*, and only individuals that were noticeably green and still physically on the alga, supposedly still feeding, were taken for imaging. With the small sea slugs *E. timida*, *E. viridis*, and *P. dendritica*, the samples were prepared simply by decapitating the animal with a razor blade and then pinning the rest of the body between the microscope slide and the cover glass. After decapitating the larger sea slug *E. crispata*, pieces of their left and right parapodia were cut out and placed on the slides for microscopy.

Images of the chloroplasts from different organisms were taken with a confocal microscope of a Mica Microhub (Leica Microsystems, Wetzlar, Germany). The objective was HC PL APO 63×/1.20 W CS2 motCORR, and the refractive index of water (1.33) was used for all samples. With the sea slugs, the images were always captured from the middle parts of both the left and right parapodia, or in the case of *P. dendritica*, from the left and right side of the body, including ceratas. *A. acetabulum* images were always taken along the stem of the algal cell, leaving out the tip and the basal rhizobia. The thick branches of *C. tomentosum* were dissected from the mid-section of a single branch, and the pieces containing the inner medulla and the outer cortex were imaged, focusing mainly on the cortex. Images from *B. plumosa* were taken along the filaments of the alga, avoiding the tips. The acquisition software was Leica Application Suite X (LASX, version 6.2.2.28360) that provides automated image capture optimization in combination with Mica. The sample characteristics were described as "dynamic" and "living" by turning on the corresponding settings from LASX. Chloroplasts were detected via chlorophyll *a* fluorescence by allowing LASX to determine the excitation laser wavelength (405 nm) and the appropriate emission detection range from the uploaded chlorophyll *a* absorption and emission spectra (downloaded from https://www.fpbase.org/spectra/; checked on 26.04.2024). After manual focusing, LASX was allowed to optimize all settings, including the laser intensity, using the "OneTouch" feature of Mica/

LASX. The optimization was done for each sample individually. Images were always captured with the LIGHTNING computational clearing feature on.

A minimum of three images from each biological replicate were analyzed using Fiji[81] to determine the area and circularity of the chloroplasts. Each image was inspected for regions where the chloroplasts were in focus, and these sections were cropped for further analysis. The recognition and segmentation of chloroplasts from the images was done using the StarDist plugin, originally designed for the recognition of fluorescent nuclei from microscope images[82]. The "Versatile (fluorescent nuclei)" model of StarDist with the default settings was deemed sufficiently accurate in recognizing the chloroplasts, but brightness and contrast adjustments were applied to the entire images in some cases to assist StarDist in identifying the chloroplasts. The hundreds to thousands of identified and segmented chloroplasts were then analyzed for area and circularity with Fiji. The circularity was defined as $(4\pi \times \text{Chloroplast Area})/\text{Chloroplast Perimeter}^2$, with the values ranging between 0 (no resemblance to a circle) and 1 (a perfect circle).

## Statistics and reproducibility
Sample sizes are detailed in the figure legends. Replicates for each experiment are detailed in the corresponding sections of the materials and methods and main text. All statistical analyses were done using R (v.4.2.3) in RStudio (2023.06.0 + 421). The base significance level was 0.05 for all analyses. The normality and homogeneity of variances of all data were inspected with Shapiro-Wilk and Levene's tests, respectively, before proceeding with testing statistically significant differences between groups using two-tailed Student's t-tests for pairwise comparisons, or one-way ANOVA when comparing three or more groups. Significant differences indicated by ANOVA were followed by post-hoc Tukey's test. If the data deviated from normality or the variances were not homogenous, statistical testing for differences was conducted using a non-parametric Kruskal–Wallis test, followed by post-hoc Dunn's tests when pertinent. Figures were prepared with the ggplot2 v3.4.1 package of R[83] in conjunction with Inkscape (v.1.3).

## Reporting summary
Further information on research design is available in the Nature Portfolio Reporting Summary linked to this article.

## Data availability
The source data for the figures are available at Zenodo, https://zenodo.org/records/14179375; https://doi.org/10.5281/zenodo.14179374[84].

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

## Acknowledgements
This work was supported by the European Research Council (ERC) under the European Union's Horizon 2020 research and innovation program, grant agreement no. 949880 to S.C. (https://doi.org/10.3030/949880), and by Fundação para a Ciência e Tecnologia, grants 2020.03278.CEECIND to S.C. (https://doi.org/10.54499/2020.03278.CEECIND/CP1589/CT0012), CEECIND/01434/2018 to P.C. (https://doi.org/10.54499/CEECIND/01434/2018/CP1559/CT0003), UID + LA/P/0094/2020 to CESAM, and by project CITAQUA, "Desenvolvimento do Projeto de Reforco do Polo de Aveiro (H4)", framed within Measure 10 of Investment TC-C10-i01—Hub Azul—Rede de Infraestruturas para a Economia Azul, financed by the Recovery and Resilience Plan (PRR) and supported by Fundo Azul of the Portuguese Government. H.M. received funding from the Finnish Cultural Foundation through the Foundation's Post Doc Pool. We thank Maria Inês Silva and Diogo Marçal for technical support in algae and sea slug maintenance. Daniel Alexandre and Bernardo Balseiro are thanked for providing *Ulva* sp. samples for the study.

## Author contributions
V.H.: conceptualization, formal analysis, investigation, methodology, resources, supervision, validation, visualization, writing—original draft. A.R.: investigation, formal analysis, writing—review & editing. H.M.: formal analysis, funding acquisition, investigation, methodology, visualization, writing—review & editing. E.T.: methodology, resources, writing—review & editing. P.C.: funding acquisition, resources, supervision, visualization, writing—review & editing. S.C.: funding acquisition, project administration, resources, supervision, writing—review & editing.

## Competing interests
The authors declare no competing interests.

## Ethics approval
This study was performed in accordance with EU legislation and directives concerning scientific research on animals, including the 3 R principles. Ethical approval is not required for studies conducted with non-cephalopod invertebrates. Wild algae and sea slugs were collected from non-protected marine coastal areas. Laboratory-reared specimens used in this work were maintained in optimized life-support systems to ensure animal welfare.
