## [Transparent Peer Review file · Communications Biology]

Loss of state transitions in Bryopsidales macroalgae and kleptoplastic sea slugs (Gastropoda, Sacoglossa)

Corresponding Author: Dr Vesa Havurinne

Version 0:

Reviewer comments:

Reviewer #1

(Remarks to the Author)

Review of the Manuscript by Havurinne et al.

It is an honor to have the opportunity to review this manuscript by Havurinne et al. This study addresses the underexplored photosynthetic properties of Ulvophyceae algae, demonstrating notably low NPQ levels in Bryopsidales. Furthermore, the authors investigated whether the NPQ characteristics are retained in incorporated kleptoplasts into Sacoglossan sea slugs, which feed on various Ulvophyceae algae. The results showed that in *E. timida* and *E. crispata*, which acquire chloroplasts from *Acetabularia* (a true Ulvophyceae species maintaining substantial NPQ, unlike Bryopsidales), NPQ levels rapidly decreased within the animals.

The authors proposed that these changes occur during the incorporation of chloroplasts by the sea slugs and highlighted morphological changes in chloroplasts driven by salinity fluctuations as a possible factor influencing NPQ changes. They observed that salinity treatments induced morphological alterations in chloroplasts within algal cells, and similar spherical transformations were noted in the chloroplasts within sea slugs, as quantified by chloroplast circularity.

The absence of NPQ in Bryopsidales is, to my knowledge, novel data that adds a fresh perspective to the study of photosynthetic traits in these algal groups. The changes in chloroplast properties within sea slugs were reliably demonstrated through careful comparisons under identical experimental conditions with their algal prey. This robust methodology convincingly highlights the differences in chloroplast functionality between the algae and the sea slugs, *E. timida*.

Although the direct connection between chloroplast morphology and NPQ remains unclear, and the physiological similarities between salinity-induced changes in algae and those in sea slugs require further investigation, these points are appropriately discussed in the manuscript.

There have been multiple observations regarding chloroplast morphological changes in kleptoplasty. This work is the first to propose a potential link between morphological transformations of kleptoplasts and their photosynthetic functionality—a significant contribution to the field. I believe this manuscript contains sufficient data to merit publication in *Communications Biology*.

Specific Comments

<Adaptive Role and Evolutionary Context>

The adaptive role of the observed phenomena and their evolutionary implications remain unclear. True Ulvophyceae are less common prey for Sacoglossans, with most species feeding on Bryopsidales. Morphological studies suggest that ancestral Sacoglossans primarily fed on Bryopsidales, and true Ulvophyceae is apomorphic food (Christa et al., 2015; Jensen, 1997).

Additionally, spherical chloroplast transformations are clearly (or weakly) occurring in *E. viridis* and *Placida dendritica*, which feed on NPQ-lacking Bryopsidales.

The authors argue that these morphological changes are actively induced by sacoglossans, thereby prolonging the retention period of the chloroplasts. This claim is indeed plausible, as supported by their interspecies comparisons.

However, as the authors demonstrated, Bryopsidales chloroplasts inherently lack NPQ. Therefore, no evidence suggests

that the correlation between changes in sphericity and photosynthetic characteristics universally occurs in many sacoglossan species that feed on Bryopsidales.

This may imply that other factors might influence the changes in sphericity, and the observed changes in photosynthetic NPQ properties could potentially be a side effect specific to *Acetabularia*, an exceptional prey alga for Sacoglossans. While these points do not detract from the novelty or importance of the manuscript, they highlight areas where further clarity or discussion could enhance the overall impact:

<Confocal microscopic studies>

As the authors also discuss, there are many differences between the changes observed in chloroplasts under different salinity conditions and the actual changes in kleptoplasty. For example, the chloroplast size decreases during kleptoplasty. This point may need to be emphasized more clearly in the Results section. Image analysis data on chloroplast size would further strengthen the discussion. I could not find information on how much time had passed since feeding for the sea slug (mainly *P. dendritica*, *E. viridis*) samples used in the observations.

<L56: Introduction of NPQ>

The term NPQ appears for the first time at this point. Given the journal's broad audience and the likely interest of researchers from various disciplines, a clear explanation of NPQ, the xanthophyll cycle, and State 1/2 transitions earlier in the manuscript would be helpful. Including a schematic diagram as supplementary data to clarify these relationships could improve accessibility for malacologists.

<Fig. 4f; L267-L277>

The description of NaF treatment is somewhat unclear. It appears NaF was applied to algae before the feeding. Did the sea slugs themselves remain exposed to NaF post-contact? Is the 135-minute incubation (without NaF?) appropriate, given NaF's metabolic properties? While I understand the inherent challenges of aligning experimental timescales for algae and sea slugs, clarifying differences in experimental conditions and the reliability of the data would be beneficial. Could differences in NaF permeability between algae and animal cells affect its impact on chloroplasts within the sea slugs? Could NaF be metabolized or diluted, rendering it ineffective within *E. timida*?

<Minar comment?>

L485-L499: Previous Studies and Results

The differences between prior studies and the present findings regarding wavelength-specific NPQ responses are intriguing. Including these points in the Results section may enhance the manuscript.

Reviewer #2

(Remarks to the Author)

The article "Evolution and theft: loss of state transitions in Bryopsidales macroalgae and photosynthetic sea slugs" uses classic photophysiological approaches to demonstrate changes in photoprotective mechanisms of green macroalgae (Bryopsidales and Ulvophyceae) plastids before and after sequestration by kleptoplastidic sea slugs. This manuscript shows for the first time that kleptoplastids in sea slugs do not retain state transition capacity and provide evidence that this may be due to changes in plastid structure during sequestration. The authors also increase our understanding of the number of species that lack state transitions in Bryopsidales algae.

Overall this is a very interesting and well constructed manuscript. The data are presented clearly and the figures are of high quality with helpful illustrations. The results are important, and the main observations are supported nicely with ancillary experiments. I enjoyed reading this and think that this is a meaningful and novel contributions to our understanding of state transitions in Bryopsidales and kleptoplastid maintenance and photophysiology in sea slugs.

I only have minor suggestions and edits:

Line 49: Change "venues" to "avenues"

Lines 53-54: delete "making their association with the sea slugs a functionally kleptoplastic one"

Line 55: delete "simply"

Line 57: delete: "quite the opposite"; Change "The..." to "Rather, the..."

Line 62: change "Furthermore" to "Further"

Line 70: Delete "in terms of their longevity"

Line 71: change "differing" to "contrasting"; change "in different" to "among"

Line 73: add "different" before "kleptoplastids" and delete "originating from different algae"

Line 74: delete "one sea slug species"; change "showed" to "revealed"; delete "having"; change "kleptoplastids" to "plastids"

Lines 74-75: delete "compared to"; change "kleptoplastids" to "plastids"; add commas around "with innate qE and xanthophyll cycle" (also, xanthophyll is misspelled)

Line 75: replace "is associated with an almost doubled kleptoplast lifetime" with "are retained almost double the time";

Lines 77-96: try to condense this a little.

Line 98: delete "functionally"

Lines 207-208: move "more oxidized" before "Qa"

Reviewer #3

(Remarks to the Author)

In their article "Evolution and theft: loss of state transitions in Bryopsidales macroalgae and photosynthetic sea slugs" Havurinne et al present extremely high quality work that really moves the field forward on a very exciting topic: kleptoplasty in sacoglossan slugs, particularly focussing on the photophysiological implications.

The discovery of a lack of state transition in plastids when they reside in the animals is highly novel and key for our understanding of this phenomenon. Overall, the study takes a broad stroke across the various different "donor algae" that the slugs feed on to obtain their stolen plastids.

I have only a few remarks:

I think it would be highly relevant for the reader to illustrate the phylogenetic relationships (by adding a cladogram, e.g. to figure 1) between the algae.

It would be good to write the number of replicates ("n=") in each of the graphs

In Figure 6a and 6b: where are we exactly in the algae and slugs (i.e. which parts of the Macro-body where imaged; the rim of the parapodia in case of the slugs)? Similarly in 7a and 7d?

Version 1:

Reviewer comments:

Reviewer #1

(Remarks to the Author)

Reviewer Comments

I am very pleased to have the opportunity to review the revised version of this manuscript. Overall, the authors have responded thoughtfully and appropriately to my previous comments, and have provided clear clarifications where misunderstandings may have occurred. The revisions have substantially improved the clarity and overall quality of the manuscript. Most of my initial concerns have been resolved, and I believe the manuscript is now much strengthened. However, a few issues remain that may benefit from further clarification.

Additionally, the sentence at the Discussion section (Line 292) appears to be abruptly cut off and should be completed for clarity and completeness.

Below, I provide my responses to the authors' numbered replies.

Answer 1

I now understand the authors' conceptual distinction between NPQ and state transitions.

I would appreciate confirmation as to whether this distinction is consistently maintained throughout the manuscript. For example, is the distinction clearly observed in Lines 348–364, where prior literature is discussed? Do references 21, 22, and 30 also employ the term "NPQ" with the same conceptual separation as defined by the authors?

As some of these points lie outside my immediate area of expertise, and given that the authors are specialists in this field, I would respectfully defer to their judgment and kindly suggest verifying whether the distinction is applied consistently throughout the manuscript.

Answer 2

The revised text in Lines 45–81 has improved the clarity of the comparison with previous studies and highlights the authors' focus on state transitions more effectively. I appreciate this improvement.

Answer 3

I would like to thank the authors for the thoughtful addition of evolutionary context in Lines 284–292. This section is intellectually stimulating and enhances the conceptual significance of the manuscript.

That said, I have a few suggestions for improvement:

Line 284: The phrase "Interestingly, the shell-bearing, functionally non-kleptoplastic ancestral sacoglossans" may be misinterpreted as referring to an extinct ancestral species. I recommend rephrasing to clarify the evolutionary implication. For example: "Interestingly, the shell-bearing sacoglossans (Oxynoacea), which have retained ancestral characteristics including the absence of functional kleptoplasty, feed exclusively on Bryopsidales algae such as Halimeda or Caulerpa."

Line 290 – use of "protokleptoplast": This appears to be a novel term coined by the authors. The concept—referring to

plastids preceding the evolution of fully functional kleptoplasts—is intriguing. However, in the absence of a clear definition, the term could be misinterpreted physiologically, for example as referring to plastids temporarily residing in the gut lumen before integration into host control. I suggest that either a more conventional term such as chloroplast or plastid be used, or that “protokleptoplast” be retained with a precise definition clearly introduced in the manuscript.

Lines 288–292: The authors suggest that the lack of complex photoprotective and acclimatory mechanisms in Bryopsidales plastids may have facilitated their early adoption by kleptoplastic sacoglossans. I would appreciate further clarification here. Is the implication that these plastids were more readily incorporated (e.g., less damaging to the host), or that they were easier to functionally maintain once incorporated? I presume the latter, but clarification would be helpful. A naive interpretation might suggest that plastids lacking photoprotection would be less stable inside the host. However, the authors’ data imply that Bryopsidales plastids may be more compatible with long-term retention precisely because their photosynthesis is less dependent on complex regulatory mechanisms like state transitions, which likely become non-functional after incorporation. If this is indeed the authors’ intended interpretation, I would suggest making it more explicit, perhaps toward the end of the Discussion, to highlight its conceptual impact.

Answers 4 and 5

The authors’ responses to these points were clear and persuasive. With the revisions made, my previous concerns have been fully resolved.

Answer 6

Understood. I look forward to future studies by the authors addressing plastid size through 3D imaging approaches.

Answer 7

No further concerns. Thank you for the clarification.

Answer 8

Thank you for the updated figure. It significantly improves clarity and contributes to the accessibility of the data.

Answers 9–10

The revisions have clarified the experimental design and strengthened the logical basis for the authors’ interpretations. I appreciate the thorough and detailed explanation.

Answer 12

Thank you for addressing this point. My initial concern stemmed from ambiguity in the description of the experimental timeline, particularly regarding potential NaF-free incubation periods. The revised manuscript resolves this issue. The authors’ discussion of NaF permeability is also appropriate and adds further depth to the experimental rationale.

Answer 13

Thank you for the clarification. I understand that physiological variability due to cultivation conditions is common in such studies.

Would the authors consider adding a brief note in the legend of Supplementary Figure 5 to indicate that differences in handling or fixation methods may have contributed to the discrepancies observed when compared to previous studies?

Reviewer #3

(Remarks to the Author)

The authors have adequately tackled all of my remarks. I have no further comments and congratulate them on a nice manuscript.

Response to reviewers

General comment by the authors: We thank all reviewers for their valuable input, taking the comments into account has improved the manuscript considerably. All major changes in the text have been now highlighted in yellow. Please find the detailed answers related to specific comments below. All specified line numbers relate to the new text version.

Reviewer #1 (Remarks to the Author):

Review of the Manuscript by Havurinne et al.

It is an honor to have the opportunity to review this manuscript by Havurinne et al. This study addresses the underexplored photosynthetic properties of Ulvophyceae algae, demonstrating notably low NPQ levels in Bryopsidales. Furthermore, the authors investigated whether the NPQ characteristics are retained in incorporated kleptoplasts into Sacoglossan sea slugs, which feed on various Ulvophyceae algae. The results showed that in *E. timida* and *E. crispata*, which acquire chloroplasts from *Acetabularia* (a true Ulvophyceae species maintaining substantial NPQ, unlike Bryopsidales), NPQ levels rapidly decreased within the animals.

Answer 1: We thank the reviewer for the generous comments. Likewise, it was an honor to receive them. We apologize if our assumptions are incorrect, but it seems that there is a mix up (at least in terminology) between NPQ and state transitions that is pertinent to clarify in order to make improvements to the manuscript according to the reviewer's suggestions. Although NPQ and state transitions are intertwined (qT being a component of NPQ related to state transitions), we have purposefully separated the two in our text to pinpoint that we are more focused on state transitions as a phenomenon itself and have left out their possible involvement in the NPQ kinetics in the algae and the sea slugs.

The authors proposed that these changes occur during the incorporation of chloroplasts by the sea slugs and highlighted morphological changes in chloroplasts driven by salinity fluctuations as a possible factor influencing NPQ changes. They observed that salinity treatments induced morphological alterations in chloroplasts within algal cells, and similar spherical transformations were noted in the chloroplasts within sea slugs, as quantified by chloroplast circularity.

The absence of NPQ in Bryopsidales is, to my knowledge, novel data that adds a fresh perspective to the study of photosynthetic traits in these algal groups.

Answer 2: Assuming that the reviewer means state transitions (re-distribution of light energy between PSII and PSI) instead of NPQ, this is correct. The lack of qE type NPQ and the xanthophyll cycle in Bryopsidales has, however, been described previously (see Christa et al. (2017) Photoprotection in a monophyletic branch of chlorophyte algae is independent of energy dependent quenching (qE). *New Phytol.* 214: 1132–1144).

The changes in chloroplast properties within sea slugs were reliably demonstrated through careful comparisons under identical experimental conditions with their algal prey. This robust

methodology convincingly highlights the differences in chloroplast functionality between the algae and the sea slugs, *E. timida*.

Although the direct connection between chloroplast morphology and NPQ remains unclear, and the physiological similarities between salinity-induced changes in algae and those in sea slugs require further investigation, these points are appropriately discussed in the manuscript. There have been multiple observations regarding chloroplast morphological changes in kleptoplasty. This work is the first to propose a potential link between morphological transformations of kleptoplasts and their photosynthetic functionality—a significant contribution to the field. I believe this manuscript contains sufficient data to merit publication in *Communications Biology*.

Specific Comments

The adaptive role of the observed phenomena and their evolutionary implications remain unclear.

True Ulvophyceae are less common prey for Sacoglossans, with most species feeding on Bryopsidales. Morphological studies suggest that ancestral Sacoglossans primarily fed on Bryopsidales, and true Ulvophyceae is apomorphic food (Christa et al., 2015; Jensen, 1997).

Answer 3: We thank the reviewer for this nice suggestion on the ancestral feeding strategy of sacoglossans. We have slightly expanded on this point in the Discussion (lines 284-292) using the suggested references.

Additionally, spherical chloroplast transformations are clearly (or weakly) occurring in *E. viridis* and *Placida dendritica*, which feed on NPQ-lacking Bryopsidales.

The authors argue that these morphological changes are actively induced by sacoglossans, thereby prolonging the retention period of the chloroplasts. This claim is indeed plausible, as supported by their interspecies comparisons.

However, as the authors demonstrated, Bryopsidales chloroplasts inherently lack NPQ. Therefore, no evidence suggests that the correlation between changes in sphericity and photosynthetic characteristics universally occurs in many sacoglossan species that feed on Bryopsidales.

Answer 4: We agree with the reviewer in that the evidence pointing to huge changes to the photosynthetic machinery in Bryopsidales during kleptoplasty is not as clear as in true ulvophytes like *Acetabularia*, where we showed also the increase in photosynthetic electron transfer (ETR) during incorporation as kleptoplasts in *E. timida*. However, even *Codium* derived kleptoplasts have been shown to exhibit higher ETR in the sea slug *E. viridis* compared to the alga itself, suggesting that similar changes may be taking place also in Bryopsidales kleptoplasts during incorporation (Serôdio et al. 2014 *Photophysiology of kleptoplasts: photosynthetic use of light by chloroplasts living in animal cells. Philos. Trans. R. Soc. B. 369: 2013024*). We have tried to emphasize this point in the Discussion (lines 349-352) of the revised manuscript, while admitting that the differences in ETR can be quite small at times, as was the case when *E. viridis* and *C.*

tomentosum were compared more recently in Morelli et al. 2024 Food shaped photosynthesis: Photophysiology of the sea slug *Elysia viridis* fed with two alternative chloroplast donors [version 2; peer review: 2 approved]. Open Res. Europe 3, 107; 10.12688/openreseurope.16162.2.

This may imply that other factors might influence the changes in sphericity, and the observed changes in photosynthetic NPQ properties could potentially be a side effect specific to *Acetabularia*, an exceptional prey alga for Sacoglossans.

Answer 5: We completely agree that the changes we noticed in NPQ and state transitions in *Acetabularia* kleptoplasts in *E. timida* are a side effect of something dramatic, and not the cause behind the changes in sphericity. We have made a small addition (the words “as a side effect”) to line 390 of Discussion.

While these points do not detract from the novelty or importance of the manuscript, they highlight areas where further clarity or discussion could enhance the overall impact:

As the authors also discuss, there are many differences between the changes observed in chloroplasts under different salinity conditions and the actual changes in kleptoplasty. For example, the chloroplast size decreases during kleptoplasty. This point may need to be emphasized more clearly in the Results section. Image analysis data on chloroplast size would further strengthen the discussion.

Answer 6: We have purposefully left the chloroplast size comparisons to less emphasis, because true size comparisons would require more detailed 3D imaging of the chloroplasts. We can't confidently say anything too detailed about the size (i.e. volume) based on the present confocal data that is always taken from only one level as a 2D image, and therefore only the most drastic changes in size that we did observe (i.e. between *Bryopsis* and *E. crispata*) are shown in the main figures of the main text. We are glad to inform, however, that 3D FIB-SEM imaging of the chloro-/kleptoplasts is currently being carried out with our collaborators. We would therefore prefer to leave the text as it is regarding the size comparison.

I could not find information on how much time had passed since feeding for the sea slug (mainly *P. dendritica*, *E. viridis*) samples used in the observations.

Answer 7: *E. viridis* and *P. dendritica* were allowed to continuously feed and only slugs that were on the surface of *Codium* (supposedly feeding) were taken for the imaging. The information, along with additional details about the feeding of the slugs prior to imaging, has been added to the Methods lines 599-602.

The term NPQ appears for the first time at this point. Given the journal's broad audience and the likely interest of researchers from various disciplines, a clear explanation of NPQ, the xanthophyll cycle, and State 1/2 transitions earlier in the manuscript would be helpful. Including a schematic diagram as supplementary data to clarify these relationships could improve accessibility for malacologists.

Answer 8: The term NPQ appears for the first time in the third paragraph of the Introduction (line 50). We agree that the NPQ processes are complex, and we have now included schematics describing qE type NPQ, xanthophyll cycle and state transitions as new Supplementary Figures 1 and 2 in the Introduction (referred to in lines 53 and 76).

The description of NaF treatment is somewhat unclear. It appears NaF was applied to algae before the feeding.

Answer 9: Correct, at time point 10 min of red light treatment of the algae.

Did the sea slugs themselves remain exposed to NaF post-contact?

Answer 10: Yes, the slugs were placed to the container containing the algae +NaF.

Is the 135-minute incubation (without NaF?) appropriate, given NaF's metabolic properties?

Answer 11: The entire experiment lasted 135 min. The maximum time the algae were exposed to NaF was 125 min (NaF added only after 10 min exposure to red light). The maximum time the slugs were exposed to NaF was 120 min (slugs added to feed on the NaF exposed algae after 15 min). However, not all the slugs were exposed to NaF for the maximum time, as we constantly observed their color change to green due to feeding, and every time we were able to find 3-4 individuals that were noticeably green, those slugs were picked and ground to a fine powder in liquid nitrogen for the 77K fluorescence measurements. The first slugs were picked out as soon as approximately 20 min after they were allowed to feed, whereas the last ones were picked after 120 min after feeding started. The schematic in Figure 4e has been modified and the experimental design has now been also clarified in the Methods lines 501-512, see also the slight changes to Fig. 4 legend.

While I understand the inherent challenges of aligning experimental timescales for algae and sea slugs, clarifying differences in experimental conditions and the reliability of the data would be beneficial. Could differences in NaF permeability between algae and animal cells affect its impact on chloroplasts within the sea slugs? Could NaF be metabolized or diluted, rendering it ineffective within *E. timida*?

Answer 12: The reason why NaF was first given to the algae themselves under red light was to lock them in state 2, so we could quite confidently say that the algae chloroplasts should be in state 2 (based on our experiment with *Acetabularia* itself using +NaF and red and far red lights; Figure 4d). We then kept them under red light and NaF while the slugs were feeding (also in the presence of NaF, as the slugs were simply put in the same container as the algae + NaF). Whether the slugs dilute and metabolize NaF themselves is unclear, and thus we cannot rule with absolute certainty out the possibility that fast de-phosphorylation inside slugs removes state 2. However, due to the small variation in 77K among NaF-treated slugs (even though their different times under the influence of NaF), the very clear absence of state 2 in slugs and almost identical results with and without NaF, we think that this possibility is small. We have tried to clarify the text and also mention these flaws in the Discussion lines 371-377.

L485-L499: Previous Studies and Results

The differences between prior studies and the present findings regarding wavelength-specific NPQ responses are intriguing. Including these points in the Results section may enhance the manuscript.

Answer 13: We agree with the reviewer, and we have now moved the Supplementary Fig. 5 (originally Suppl. Fig. 7) to the Results section lines 182-186. However, the experiments with multi-color PAM were done using *Acetabularia* and *Elysia timida* grown at the university of Turku, Finland, a long time ago, and the growth conditions were slightly different than what were used in the main result figures of the main text. Also, as mentioned in the Supplementary Figure legend, the fixation method of the slugs and algae was 1% alginate that requires the use of CaCl_2 , which might affect a true comparison with the experimental setups used in, e.g. Figures 3 and 5 where photosynthesis from *Acetabularia* and *Elysia timida* was also measured with a PAM fluorometer. We would therefore prefer to maintain those Multi-Color PAM data in the supplement.

Reviewer #2 (Remarks to the Author):

The article “Evolution and theft: loss of state transitions in Bryopsidales macroalgae and photosynthetic sea slugs” uses classic photophysiological approaches to demonstrate changes in photoprotective mechanisms of green macroalgae (Bryopsidales and Ulvophyceae) plastids before and after sequestration by kleptoplastidic sea slugs. This manuscript shows for the first time that kleptoplastids in sea slugs do not retain state transition capacity and provide evidence that this may be due to changes in plastid structure during sequestration. The authors also increase our understanding of the number of species that lack state transitions in Bryopsidales algae.

Overall this is a very interesting and well constructed manuscript. The data are presented clearly and the figures are of high quality with helpful illustrations. The results are important, and the main observations are supported nicely with ancillary experiments. I enjoyed reading this and think that this is a meaningful and novel contributions to our understanding of state transitions in Bryopsidales and kleptoplastid maintenance and photophysiology in sea slugs.

Answer 14: We wholeheartedly thank the reviewer for this praise.

I only have minor suggestions and edits:

Line 49: Change “venues” to “avenues”

Lines 53-54: delete “making their association with the sea slugs a functionally kleptoplastic one”

Line 55: delete “simply”

Line 57: delete: “quite the opposite”; Change “The...” to “Rather, the...”

Line 62: change “Furthermore” to “Further”

Line 70: Delete “in terms of their longevity”

Line 71: change “differing” to “contrasting”; change “in different” to “among”

Line 73: add “different” before “kleptoplastids” and delete “originating from different algae”

Line 74: delete “one sea slug species”; change “showed” to “revealed”; delete “having”; change “kleptoplastids” to “plastids”

Lines 74-75: delete “compared to”; change “kleptoplastids” to “plastids”; add commas around “with innate qE and xanthophyll cycle” (also, xanthophyll is misspelled)

Line 75: replace “is associated with an almost doubled kleptoplast lifetime” with “are retained almost double the time”;

Lines 77-96: try to condense this a little.

Line 98: delete “functionally”

Lines 207-208: move “more oxidized” before “Qa”

Answer 15: We have modified the text according to the comments by the reviewer, please see the highlighted text parts in the Introduction (lines 45-88) and Results (lines 165-168). However, to avoid ambiguity we have purposefully not used the term “plastid” regarding chloroplasts inside sea slugs in this manuscript, and would prefer to use “chloroplasts” when referring to plastids that are still inside the algae.

Reviewer #3 (Remarks to the Author):

In their article "Evolution and theft: loss of state transitions in Bryopsidales macroalgae and photosynthetic sea slugs" Havurinne et al present extremely high quality work that really moves the field forward on a very exciting topic: kleptoplasty in sacoglossan slugs, particularly focussing on the photophysiological implications.

The discovery of a lack of state transition in plastids when they reside in the animals is highly novel and key for our understanding of this phenomenon. Overall, the study takes a broad stroke across the various different "donor algae" that the slugs feed on to obtain their stolen plastids.

Answer 16: We thank also Reviewer #3 for the uplifting comments, they were highly motivating.

I have only a few remarks:

I think it would be highly relevant for the reader to illustrate the phylogenetic relationships (by adding a cladogram, e.g. to figure 1) between the algae.

Answer 17: A cladogram has now been added to Fig. 1 as panel (a). Figure 1 legend was changed accordingly.

It would be good to write the number of replicates ("n=") in each of the graphs

Answer 18: We have now added the number of replicates to each pertinent graph and altered the figure legends accordingly.

In Figure 6a and 6b: where are we exactly in the algae and slugs (i.e. which parts of the Macro-body where imaged; the rim of the parapodia in case of the slugs?)? Similarly in 7a and 7d?

Answer 19: Only the parapodia (both left and right) were imaged from the slugs, except for *P. dendritica*, where the left and right side of the body, including the cerata, were imaged. Acetabularia were imaged along the stems of the algae, avoiding the tip and base of the cells. For Codium, a single branch was always dissected from approximately middle of the branch, and this section containing both the inner medulla and the outer cortex was imaged, focusing mainly on the outer cortex where most of the chloroplasts reside. Bryopsis were imaged all over the algal filaments, avoiding the tips of the algae. We have now added more details to the Materials and Methods lines 609-616 regarding this.

REVIEWERS' COMMENTS:

Reviewer #1 (Remarks to the Author):

Reviewer Comments

I am very pleased to have the opportunity to review the revised version of this manuscript. Overall, the authors have responded thoughtfully and appropriately to my previous comments, and have provided clear clarifications where misunderstandings may have occurred. The revisions have substantially improved the clarity and overall quality of the manuscript. Most of my initial concerns have been resolved, and I believe the manuscript is now much strengthened. However, a few issues remain that may benefit from further clarification.

Additionally, the sentence at the Discussion section (Line 292) appears to be abruptly cut off and should be completed for clarity and completeness.

Response 1: The sentence has been finished, thank you for noticing.

Below, I provide my responses to the authors' numbered replies.

Answer 1

I now understand the authors' conceptual distinction between NPQ and state transitions. I would appreciate confirmation as to whether this distinction is consistently maintained throughout the manuscript. For example, is the distinction clearly observed in Lines 348–364, where prior literature is discussed? Do references 21, 22, and 30 also employ the term “NPQ” with the same conceptual separation as defined by the authors?

As some of these points lie outside my immediate area of expertise, and given that the authors are specialists in this field, I would respectfully defer to their judgment and kindly suggest verifying whether the distinction is applied consistently throughout the manuscript.

Response 2: We have been careful in selecting the wording related to NPQ throughout the manuscript, and it is our sentiment that the phrasing is consistent with the existing literature on the topic.

Answer 2

The revised text in Lines 45–81 has improved the clarity of the comparison with previous studies and highlights the authors' focus on state transitions more effectively. I appreciate this improvement.

Answer 3

I would like to thank the authors for the thoughtful addition of evolutionary context in Lines 284–292. This section is intellectually stimulating and enhances the conceptual significance of the manuscript.

That said, I have a few suggestions for improvement:

Line 284: The phrase “Interestingly, the shell-bearing, functionally non-kleptoplastic ancestral sacoglossans” may be misinterpreted as referring to an extinct ancestral species. I recommend rephrasing to clarify the evolutionary implication. For example: “Interestingly, the shell-bearing sacoglossans (Oxynoacea), which have retained ancestral characteristics including the absence of functional kleptoplasty, feed exclusively on Bryopsidales algae such as Halimeda or Caulerpa.”

Response 3: We have modified the text according to the reviewer’s suggestion.

Line 290 – use of “protokleptoplast”: This appears to be a novel term coined by the authors. The concept—referring to plastids preceding the evolution of fully functional kleptoplasts—is intriguing. However, in the absence of a clear definition, the term could be misinterpreted physiologically, for example as referring to plastids temporarily residing in the gut lumen before integration into host control. I suggest that either a more conventional term such as chloroplast or plastid be used, or that “protokleptoplast” be retained with a precise definition clearly introduced in the manuscript.

Response 4: The term was changed to “chloroplast”.

Lines 288–292: The authors suggest that the lack of complex photoprotective and acclimatory mechanisms in Bryopsidales plastids may have facilitated their early adoption by kleptoplastic sacoglossans. I would appreciate further clarification here. Is the implication that these plastids were more readily incorporated (e.g., less damaging to the host), or that they were easier to functionally maintain once incorporated? I presume the latter, but clarification would be helpful. A naive interpretation might suggest that plastids lacking photoprotection would be less stable inside the host. However, the authors’ data imply that Bryopsidales plastids may be more compatible with long-term retention precisely because their photosynthesis is less dependent on complex regulatory mechanisms like state transitions, which likely become non-functional after incorporation. If this is indeed the authors’ intended interpretation, I would suggest making it more explicit, perhaps toward the end of the Discussion, to highlight its conceptual impact.

Response 5: We have now added a sentence towards the end of the discussion, in lines 435-438 to emphasize this point.

Answers 4 and 5

The authors’ responses to these points were clear and persuasive. With the revisions made, my previous concerns have been fully resolved.

Answer 6

Understood. I look forward to future studies by the authors addressing plastid size through 3D imaging approaches.

Answer 7

No further concerns. Thank you for the clarification.

Answer 8

Thank you for the updated figure. It significantly improves clarity and contributes to the accessibility of the data.

Answers 9–10

The revisions have clarified the experimental design and strengthened the logical basis for the authors' interpretations. I appreciate the thorough and detailed explanation.

Answer 12

Thank you for addressing this point. My initial concern stemmed from ambiguity in the description of the experimental timeline, particularly regarding potential NaF-free incubation periods. The revised manuscript resolves this issue. The authors' discussion of NaF permeability is also appropriate and adds further depth to the experimental rationale.

Answer 13

Thank you for the clarification. I understand that physiological variability due to cultivation conditions is common in such studies.

Would the authors consider adding a brief note in the legend of Supplementary Figure 5 to indicate that differences in handling or fixation methods may have contributed to the discrepancies observed when compared to previous studies?

Response 6: We have now added notes saying “Both the algae and the sea slugs were fixed in a drop of 1% alginate for the measurements, polymerized with 50 mM CaCl₂, *which is a different method of fixation to that used in the experiments of the main text figures.*” and “The algae and sea slugs used in these measurements were of the same laboratory population as used in Havurinne and Tyystjärvi (2020)³⁰, *and not the same population as the one used in the main manuscript text experiments. The differences in handling and fixation may themselves cause differences in NPQ between experiments in different studies.*” to the legend of Supplementary Figure 5 legend to emphasize this.